# Adipsin promotes bone marrow adiposity by priming mesenchymal stem cells

**Nicole Aaron[1,2], Michael J Kraakman[1,3], Qiuzhong Zhou[4], Qiongming Liu[1,5], Samantha Costa[6,7,8], Jing Yang[1,5], Longhua Liu[1,5], Lexiang Yu[1,5], Liheng Wang[1,3], Ying He[1,5], Lihong Fan[1,5], Hiroyuki Hirakawa[9,10], Lei Ding[9,10], James Lo[11], Weidong Wang[12], Baohong Zhao[13], Edward Guo[14], Lei Sun[4], Cliff J Rosen[6], Li Qiang[1,5]\***

[1]Naomi Berrie Diabetes Cente, Columbia University, New York, United States; [2]Department of Pharmacology, Columbia University, New York, United States; [3]Department of Medicine, Columbia University, New York, United States; [4]Cardiovascular and Metabolic Disorders Program, Duke-NUS Medical School, Singapore, Singapore; [5]Department of Pathology and Cellular Biology, Columbia University, New York, United States; [6]Center for Molecular Medicine, Maine Medical Center Research Institute, Scarborough, United States; [7]School of Medicine, Tufts University, Boston, United States; [8]Graduate School of Biomedical Science and Engineering, University of Maine, Orono, United States; [9]Department of Microbiology and Immunology, Columbia University, New York, United States; [10]Department of Rehabilitation and Regenerative Medicine, Vagelos College of Physicians and Surgeons, New York, United States; [11]Weill Center for Metabolic Health, Cardiovascular Research Institute, and Division of Cardiology, Weill Cornell Medical College, New York, United States; [12]Department of Medicine, Division of Endocrinology, Harold Hamm Diabetes Center, The University of Oklahoma Health Science Center, Oklahoma City, United States; [13]Arthritis and Tissue Degeneration Program and The David Z. Rosensweig Genomics Research Center, Hospital for Special Surgery, Department of Medicine, Weill Cornell Medical College; Graduate Program in Cell & Developmental Biology, Weill Cornell Graduate School of Medical Sciences, New York, United States; [14]Department of Biomedical Engineering, Columbia University, New York, United States

**\*For correspondence:**
lq2123@cumc.columbia.edu

**Competing interests:** The authors declare that no competing interests exist.

## Abstract

**Background:** Marrow adipose tissue (MAT) has been shown to be vital for regulating metabolism and maintaining skeletal homeostasis in the bone marrow (BM) niche. As a reflection of BM remodeling, MAT is highly responsive to nutrient fluctuations, hormonal changes, and metabolic disturbances such as obesity and diabetes mellitus. Expansion of MAT has also been strongly associated with bone loss in mice and humans. However, the regulation of BM plasticity remains poorly understood, as does the mechanism that links changes in marrow adiposity with bone remodeling.

**Methods:** We studied deletion of Adipsin, and its downstream effector, C3, in C57BL/6 mice as well as the bone-protected PPARγ constitutive deacetylation 2KR mice to assess BM plasticity. The mice were challenged with thiazolidinedione treatment, calorie restriction, or aging to induce bone loss and MAT expansion. Analysis of bone mineral density and marrow adiposity was performed using a μCT scanner and by RNA analysis to assess adipocyte and osteoblast markers. For *in vitro* studies, primary bone marrow stromal cells were isolated and subjected to osteoblastogenic or adipogenic differentiation or chemical treatment followed by morphological and molecular

analyses. Clinical data was obtained from samples of a previous clinical trial of fasting and high-calorie diet in healthy human volunteers.

**Results:** We show that Adipsin is the most upregulated adipokine during MAT expansion in mice and humans in a PPARγ acetylation-dependent manner. Genetic ablation of Adipsin in mice specifically inhibited MAT expansion but not peripheral adipose depots, and improved bone mass during calorie restriction, thiazolidinedione treatment, and aging. These effects were mediated through its downstream effector, complement component C3, to prime common progenitor cells toward adipogenesis rather than osteoblastogenesis through inhibiting Wnt/β-catenin signaling.

**Conclusions:** Adipsin promotes new adipocyte formation and affects skeletal remodeling in the BM niche. Our study reveals a novel mechanism whereby the BM sustains its own plasticity through paracrine and endocrine actions of a unique adipokine.

**Funding:** This work was supported by the National Institutes of Health T32DK007328 (NA), F31DK124926 (NA), R01DK121140 (JCL), R01AR068970 (BZ), R01AR071463 (BZ), R01DK112943 (LQ), R24DK092759 (CJR), and P01HL087123 (LQ).

## Introduction

The skeletal system is a dynamic organ affected by obesity and its associated comorbidities, such as type 2 diabetes mellitus (T2DM), cardiovascular diseases (CVDs), and cancer (*Tabas et al., 2010*; *Gallagher and LeRoith, 2010*; *Guh et al., 2009*). Residing within the bone marrow (BM) are a depot of adipocytes, which appear physiologically and developmentally distinct from other adipose tissues. These bone marrow adipocytes (BMAds) were long considered as inert space fillers in the marrow niche, filling up to 70% of BM volume (*Scheller et al., 2016*). However, they are now recognized as important regulators of hematopoiesis, tumorigenesis, metabolism, and skeletal remodeling (*Sheu and Cauley, 2011*; *Zhang et al., 2019*; *McDonald et al., 2017*). Marrow adipose tissue (MAT) increases with aging (*Justesen et al., 2001*), diabetes treatment by thiazolidinediones (TZDs) (*Lecka-Czernik et al., 1999*; *Gimble et al., 1996*), and, paradoxically, calorie restriction (CR) (*Devlin et al., 2010*) irrespective of species, gender, or ethnicity (*Nazare et al., 2012*; *Rosen et al., 2009*). The additional expansion of MAT has been observed in other conditions such as glucocorticoid treatment (*Vande Berg et al., 1999*; *Li et al., 2013*) and streptozotocin-induced diabetes (*Botolin and McCabe, 2007*). In fact, MAT accounts for 10% of total fat in young, healthy adult humans and can be expanded to up to 30% of total body fat (*Wang et al., 2018*; *Cawthorn et al., 2014*). These changes in MAT are frequently accompanied by significant bone loss (*Justesen et al., 2001*; *Rosen et al., 2009*). Expansion of MAT has been correlated with oxidative stress, inflammation, and changes in the metabolic environment and hormonal milieu, all potential stimuli to bone loss as well (*Shapses and Sukumar, 2012*). Indeed, our understanding of MAT is primarily gained from studying the bone-forming cell, the osteoblast, and the bone-resorbing cell, the osteoclast – both of which are directly involved in skeletal remodeling. Compared to the interplay between osteoblasts and osteoclasts, the regulation of MAT remains much less understood.

BMAds and osteoblasts originate from common bone marrow stromal cells (BMSCs) (*Dominici et al., 2006*). The balance between adipocyte and osteoblast differentiation is regulated by complex and dynamic intra- and extra-cellular factors. The nuclear receptor PPARγ is a key factor in determining this balance by driving adipogenesis while inhibiting osteoblastogenesis (*Chen et al., 2016*; *Lecka-Czernik et al., 2002*). Though they are potent insulin sensitizers (*Abbas et al., 2012*; *Lehrke and Lazar, 2005*), PPARγ agonist TZDs promote bone loss as well as MAT expansion, although the mechanism is not entirely clear (*Mannucci and Dicembrini, 2015*; *Pavlova et al., 2018*). Besides ligand-mediated agonism, PPARγ activity is modified by various post-translational modifications (PTMs) (*Hu et al., 1996*; *Iankova et al., 2006*; *Choi et al., 2010*; *Dutchak et al., 2012*; *Pascual et al., 2005*). We have previously identified two residues on PPARγ, Lys268 and Lys293, that are deacetylated by the $NAD^+$-dependent deacetylase, SirT1 (*Qiang et al., 2012*). PPARγ acetylation is commonly observed in obesity, diabetes, and aging and reduced with cold exposure and 'browning' (*Qiang et al., 2012*). Constitutive PPARγ deacetylation (K268R/K293R, 2KR) improves the metabolic phenotype of diet-induced obese (DIO) mice and protects against TZD-induced bone loss and MAT expansion (*Kraakman et al., 2018*). Thus, it is plausible that the PTMs of PPARγ are involved in regulating BM homeostasis.

Adipose tissue is recognized as an important endocrine organ due to its secretion of a variety of cytokines to regulate processes such as energy homeostasis, insulin sensitivity, inflammation, and skeletal remodeling (*Horowitz et al., 2017*). Among them, Adipsin was the first adipocyte-secreted protein discovered in 1987 (*Cook et al., 1987*). It has since been identified as Complement Factor D (CFD) (*Rosen et al., 1989*; *White et al., 1992*), a rate-limiting factor in the alternative pathway of the complement system (*Xu et al., 2001*). While many components of the complement system are produced by hepatocytes, macrophages, or endothelial cells, Adipsin is produced nearly exclusively by adipocytes (*Choy et al., 1992*) through the activation of PPARγ (*Tontonoz et al., 1994*). Despite this, the overall function of Adipsin as a secretory protein in vivo remains less well understood. Recently, Adipsin has been shown to promote insulin secretion by pancreatic β-cells and to protect β-cells from cell death (*Lo et al., 2014*; *Gómez-Banoy et al., 2019*), but to be dispensable for atherogenesis in *Ldlr*[-/-] mice (*Liu et al., 2021*). Unlike other adipokines, such as Leptin and Adiponectin, the function and regulation of Adipsin in the BM are completely unknown.

In the present study, we identified Adipsin among the most responsive factors to MAT expansion in both mice and humans. We employed various bone loss models with Adipsin loss-of-function in vivo and in vitro to systemically investigate the function and regulation of Adipsin in BM. Our findings reveal Adipsin to be an important causal mechanism between adipocytes and the BM microenvironment by influencing MAT expansion and, consequently, skeletal health in metabolic conditions such as diet restriction, T2DM, and aging.

## Materials and methods

### Key resources table

| Reagent type (species) or resource | Designation | Source or reference | Identifiers | Additional info |
|---|---|---|---|---|
| Strain, strain background (*Mus musculus*) | C57BL/6J | The Jackson Laboratory | 000664 | |
| Cell line (*Homo sapiens*) | HEK293T | ATCC | CRL-3216 | |
| Cell line (*Mus musculus*) | C3H10T1/2, Clone 8 | ATCC | CCL-226 | |
| Antibody | Mouse Complement Factor D/Adipsin (sheep polyclonal) | R&D Systems | AF5430 | 1:1000 |
| Antibody | Adiponectin (rabbit polyclonal) | ThermoFisher Scientific | PA1-054 | 1:1000 |
| Antibody | C3/C3b/C3c (rabbit polyclonal) | Proteintech | 21337-1-AP | 1:1000 |
| Antibody | FABP4 (aP2) (rabbit polyclonal) | Cell Signaling Technology | 2120 | 1:1000 |
| Antibody | HSP90 (rabbit polyclonal) | Proteintech | 1371-1-AP | 1:1000 |
| Antibody | Phospho-GSK-3β (rabbit polyclonal) | Cell Signaling Technology | 9336 | 1:500 |
| Antibody | β-Catenin (rabbit polyclonal) | Cell Signaling Technology | 9562 | 1:500 |
| Peptide, recombinant protein | Factor D Protein, Mouse, Recombinant (His Tag) | Sino Biological | 50539-M08H | 1 μg/mL |
| Chemical compound, drug | SB 290157 trifluoroacetate | R&D Systems | 6860/5 | 1 μM |
| Chemical compound, drug | Alizarin Red Stain | Sigma-Aldrich | A5533 | |
| Chemical compound, drug | Oil Red O Stain | Electron Microscopy Sciences | 26079 | |
| Chemical compound, drug | Osmium tetroxide | Polysciences | 23310 | |
| Software, algorithm | Quantum FX μCT Scanner | Perkin-Elmer | CLS149276 | |
| Software, algorithm | Analyze 12.0 | Analyze Direct | N/A | |

### Animal studies

The generation of 2KR mice on a C57BL/6 genetic background was described previously (*Kraakman et al., 2018*). The Adipsin KO mouse line was obtained from Dr. James Lo (Weill Cornell Medical College). The mouse lines of C3 KO (Stock No: 029661), *Adipoq-Cre* (Stock No: 028020), and *Pparg*[loxp] (Stock No: 004584) were purchased from Jackson Laboratory. All the lines were bred and maintained on the C57BL/6J background. Mice were housed at room temperature (RT, 23 ± 1° C) on a 12 hr light/12 hr dark cycle with access to food and water ad libitum. The High Fat Diet (HFD) contained 60% calories from fat, 20% from protein, and 20% from carbohydrates (Research Diets: D12492). Rosi maleate (Avandia) (Abcam, ab142461) was mixed into HFD at 100 mg/kg at

Research Diets (New Brunswick, NJ) to achieve a dose of approximately 5 mg/kg body weight (BW). In the CR diet, mice received 70% of their daily food intake (chow) twice a day for 4 weeks. For the glucose tolerance test (GTT), mice were fasted overnight in cages with fresh bedding and intraperitoneally (i.p.) injected with glucose (2 g/kg BW). Blood glucose was measured with a Breeze2 glucometer (Bayer) at indicated time points. For the insulin tolerance test (ITT), mice were fasted for 4 hr and injected i.p. with insulin (0.75 U insulin/kg BW). Body compositions were determined by EchoMRI. All animal protocols used in this study were reviewed and approved by the Columbia University Animal Care and Utilization Committee.

## Sequencing analysis

We downloaded the gene expression dataset of whole-tissue BM in mice fed ad libitum or on CR for 3 weeks from GEO (GSE124063) (*Collins et al., 2019*). Differentially expressed genes (DEGs) were assessed by a fold change of 1 in the gene expression difference between control and CR. The secreted proteins were obtained from the UniProt with the keyword Secreted [KW-0964] (*Bateman and UniProt Consortium, 2019*). Only the manually annotated secretin proteins were retained for further analysis and experiments. We employed DAVID 6.8 to perform the functional annotation (Cellular Component [CC]) for the DEGs (*Huang et al., 2009*). The enriched CC was evaluated using a false discovery rate (FDR, Benjamini–Hochberg method) of 0.1 and DEG number $\geq$ 10.

## Bone processing and analysis

Femurs were collected and fixed in 10% neutral buffered formalin overnight at 4°C, and subsequently used for bone micro-architecture analysis and lipid quantification. For micro-architecture analysis, a Quantum FX μCT Scanner (Perkin-Elmer) was used for scanning. For lipid quantification, a 14% EDTA solution was used to decalcify bone for at least 2 weeks with frequent changes. The bones were then stained for 48 hr in a 1% osmium tetroxide, 2.5% potassium dichromate solution at RT, washed in tap water for at least 2 hr, and imaged by μCT. The software Analyze 12.0 was used to quantify lipid volume and μCT scan parameters and performed according to their bone micro-architecture add-on. Marrow adipose sections were determined based on consistent 250 slice intervals measured from the identified growth plate of the femur.

## BMSC isolation

BMSCs were isolated from 4-week-old mice. Briefly, the mice were euthanized, and bones were dissected and washed in sterile PBS. BM was flushed out using a 25-gauge needle into α modification of Minimum Essential Medium (αMEM) supplemented with 10% fetal bovine serum (FBS) and 1% penicillin/streptomycin. Upon reaching 70–80% confluence, BMSCs were passaged and plated in 6-well or 12-well plates for experiments.

## Adipose stromal cell (ASC) isolation

Inguinal fat pads with the lymph nodes removed were dissected from 5- to 6-week-old wild-type (WT) and Adipsin knock-out (KO) mice, followed by mincing and digesting in Liberase (Sigma-Aldrich, catalog # 5401127001) at 37°C for 30 min with gentle agitation. After passing through a 100 μM pore cell strainer, the ASC was pelleted by centrifuging at 400 g for 5 min at 4°C. The pellet was resuspended and plated in basic medium (DMEM supplemented with 10% FBS, 1% penicillin/streptomycin, 1% gentamycin). Upon reaching 70–80% confluence, ASCs were passaged and plated in 6-well plates for experiments.

## Cell lines

HEK293T (ATCC CRL-3216) and C3H10T1/2 (ATCC CCL-226) were obtained from ATCC along with thorough reports of analysis confirming the cell line identification, authentication, and confirmed negative testing for mycoplasma contamination. The PPARγ cDNAs, including PPARγ2-WT and PPARγ2-2KR, were cloned into a doxycycline-inducible lentiviral plasmid, pTRIPZ (Thermo Open Biosystems) (*Li et al., 2016*), and were stably overexpressed into *PPARγ*$^{-/-}$ mouse embryonic fibroblasts (MEFs) (*Rosen et al., 2002*) with a selection of puromycin (2.5 μg/mL). Cells were grown in high-glucose DMEM (Corning, 10-017) supplemented with 10% FBS (heat inactivated; Corning, 35-011-CV)

and 1% penicillin/streptomycin. These PPARγ cell lines have been previously validated and tested (*Kraakman et al., 2018*).

## Cell culture treatment

To induce osteogenic differentiation, we used mineralization-inducing αMEM containing 100 μM/mL ascorbic acid and 2 mM β-glycerophosphate. Two days after induction, the cells were maintained in the same medium changed every 2–3 days. Approximately 50 mM Alizarin Red S (Sigma-Aldrich) in water was used to stain calcium deposition after 21 days. To induce adipogenic differentiation, we cultured BMSCs in DMEM containing 1 μM dexamethasone, 0.5 mM 3-isobutyl-1-methylxanthine, 10 μg/mL insulin, and 5 μM rosiglitazone (Rosi). Two days after induction, the cells were maintained in medium containing 10 μg/mL insulin and 5 μM Rosi until fully differentiated. Media were changed every 2–3 days. Recombinant Adipsin (Sino Biological, 1 μg/mL) and C3aR inhibitor, SB290157 (R&D Systems, 1 μM), were added to the medium 2 days before induction and included in the maintenance media when necessary. Recombinant Adipsin, SB290157, and recombinant Wnt3a (R&D Biosystems, 20 ng/mL) were added to the media of C3H10T1/2 cells 24 hr before harvesting. Oil Red O staining was performed to detect the lipid droplets using an Oil Red O staining kit according to the manufacturer's instruction (Electron Microscopy Sciences).

## Quantitative real-time PCR (qPCR)

Tissues and cells were lysed with 1 mL TRI reagent (Sigma-Aldrich). After phase separation through the addition of 250 μL chloroform, RNA was isolated using the NucleoSpin RNA Kit (Macherey-Nagel, Inc). The High-Capacity cDNA Reverse Transcription Kit (Applied Biosystems) was used to synthesize cDNA from 1 μg total RNA. qPCR was performed on a Bio-Rad CFX96 Real-Time PCR system with the GoTaq qPCR Master Mix (Promega). Relative gene expression levels were calculated using the ΔΔCt method with CPA as the reference gene.

## Western blots

Cells were lysed and tissues were homogenized by Polytron homogenizer immediately after dissection in western extraction buffer (150 mmol/L NaCl, 10% glycerol, 1% NP-40, 1 mmol/L EDTA, 20 mmol/L NaF, 30 mmol/L sodium pyrophosphate, 0.5% sodium deoxycholate, 0.05% SDS, 25 mmol/L Tris-HCl: pH 7.4) containing protease inhibitor cocktail (Roche). The lysate was then sonicated, and debris was removed by centrifugation. SDS-PAGE and western blotting were performed and detected with ECL (Thermo Scientific). Antibodies used for western blot analysis were as follows: anti-Adipsin (R&D Systems, catalog# AF5430), anti-adiponectin (Invitrogen, catalog# PA1-054), anti-C3 (Proteintech, catalog# 21337-1-AP), anti-FABP4 (Cell Signaling Technology, catalog# 2120), anti-phospho-GSK-3β (Cell Signaling Technology, catalog# 9336), anti-HSP90 (Proteintech Group, Inc, catalog# 1371-1-AP), and anti-β-Catenin (Cell Signaling Technology, catalog# 9562).

## Immunohistochemistry

Femurs were collected and fixed in 10% neutral buffered formalin overnight at 4˚C and kept long term in 70% ethanol at 4˚C. They were subsequently decalcified in a 20% EDTA solution for 4 weeks before becoming embedded with paraffin. Allocated sections were stained with Harris hematoxylin and eosin (H&E). Immunohistochemistry for β-Catenin (Cell Signaling Technology, catalog# 9562) was performed using a 1:100 dilution in PBST.

## Human MAT studies

BMAd RNA-seq analysis was performed on samples from a previous clinical trial of fasting and high-calorie diet in healthy human volunteers (*Fazeli et al., 2021*). The study was approved by the Partners HealthCare Institutional Review Board and complied with the Health Insurance Portability and Accountability Act guidelines. Written informed consent was obtained from all subjects. BMAds from those volunteers were separated by centrifugation from marrow sera and the RNA was extracted and frozen. RNA-seq analysis was performed at the Vermont Institute of Genomics Research. Samples were de-identified, and the analysis was blinded to assignment. Data analyses were performed using STRING (Search Tool for the Retrieval of Interacting Genes/Proteins). Protein

coding genes were assessed with p<0.05, FDR < 0.05. Morpheus software (Broad Institute) was also utilized with hierarchical clustering done by Euclidean distance.

## Statistics

Values are presented as mean ± SEM. We used unpaired two-tailed Student's *t*-test and two-way ANOVA to evaluate statistical significance with a p<0.05 considered to be statistically significant.

## Results

### Adipsin is robustly induced in the BM during MAT expansion

CR is a broadly adapted intervention for metabolic improvements. It induces the shrinkage of most adipose tissue depots, but, paradoxically, expands MAT as in the femur (*Figure 1A*). Transcriptomic analyses of whole BM revealed that the most pronounced changes during CR are associated with the secretome (*Figure 1—figure supplement 1A*). Interestingly, *Cfd* (encoding Adipsin) is among the top induced genes by CR in the BM and is the most abundant among all secretory genes (*Figure 1B*). Adipsin is a highly expressed adipokine in peripheral fat pads, such as subcutaneous white adipose tissue (SWAT) and epididymal white adipose tissue (EWAT) (*Min and Spiegelman, 1986*). However, its expression and regulation have not been examined in MAT. qPCR analysis validated an approximately ninefold induction of *Cfd* in the BM by CR compared to ad libitum-fed mice, accompanied by increased adipocyte genes (*Pparg2*, *Fabp4*, *Plin1*) and decreased osteoblast markers (*Osteocalcin*, *Col1a1*) (*Figure 1C*), whereas its induction in EWAT and SWAT was milder or unchanged, respectively (*Figure 1—figure supplement 1B, C*). In the bone, protein levels of Adipsin are also notably increased after CR (*Figure 1D*). Of note, CR increases circulating Adiponectin levels, attributed to its production by BMAds (*Cawthorn et al., 2014*). However, the upregulation of BM Adipsin, even though it is more pronounced and at a higher abundance than Adiponectin, is unlikely to contribute significantly to the circulating levels, as plasma Adipsin was not significantly increased in CR mice (*Figure 1D, E*). This data suggests that, unlike other adipokines, Adipsin is uniquely regulated in the BM, highlighting its potential role in regulating the BM microenvironment.

To examine whether the induction of BM Adipsin is restricted to CR, we employed a distinct model to induce MAT expansion in vivo using a PPARγ agonist (TZD) known as Rosi in DIO mice. This condition could induce profound MAT expansion, as indicated by a significant increase of osmium tetroxide staining of lipid droplets (*Figure 1F*). In this model, *Cfd* expression in the BM was upregulated >12-fold, more than other adipocyte markers (e.g., *Pparg2* and *Fabp4*) and Adiponectin (*Adipoq*) (*Figure 1G*). Increased expression of Perilipin 1 (*Plin1*) was also robustly induced by Rosi treatment in the BM, indicating abundant MAT expansion (*Styner et al., 2017*; *Li et al., 2019*). In contrast, the upregulation of *Cfd* expression occurred to a lesser extent in the EWAT and SWAT (*Figure 1—figure supplement 1D, E*). The concentrations of bone and plasma Adipsin were elevated only moderately, differing from a significant increase in Adiponectin levels (*Figure 1H, I*). Again, Adipsin appears to be the most responsive adipokine to MAT expansion in the BM specifically.

MAT expansion is generally believed to occur concurrently with bone loss. However, a short 3-week Rosi treatment was sufficient to stimulate a fivefold expansion of MAT in the femurs (*Figure 1J*). In contrast, neither the trabecular nor the cortical bone mineral density (BMD) was decreased by short-term Rosi treatment (*Figure 1K, L*). This data suggests that BM adipogenesis precedes the significant bone loss by TZD and that BMAd-derived factors, such as Adipsin, may participate in regulating downstream changes in the bone.

### Ablation of Adipsin inhibits BM adipogenesis and protects bone

To obtain direct evidence of that Adipsin could influence BM plasticity, we utilized Adipsin KO mice. Upon CR, Adipsin KO mice displayed significantly lower bone marrow adiposity (BMA) detected by osmium tetroxide staining for lipid droplets (*Figure 2A, B*), though H&E staining revealed minimal lipid droplet accumulation in both CR conditions and chow-fed controls (*Figure 2—figure supplement 1E*). The inhibited MAT was underlined by decreased expression of adipogenesis makers in the BM, including *Pparg*, *Adipoq*, *Fabp4*, *Fasn*, *Gdp1*, and *Perilipin* (*Figure 2C*). Interestingly, in contrast to their reduced BMA, Adipsin KO mice preserved more fat mass, despite the similar response

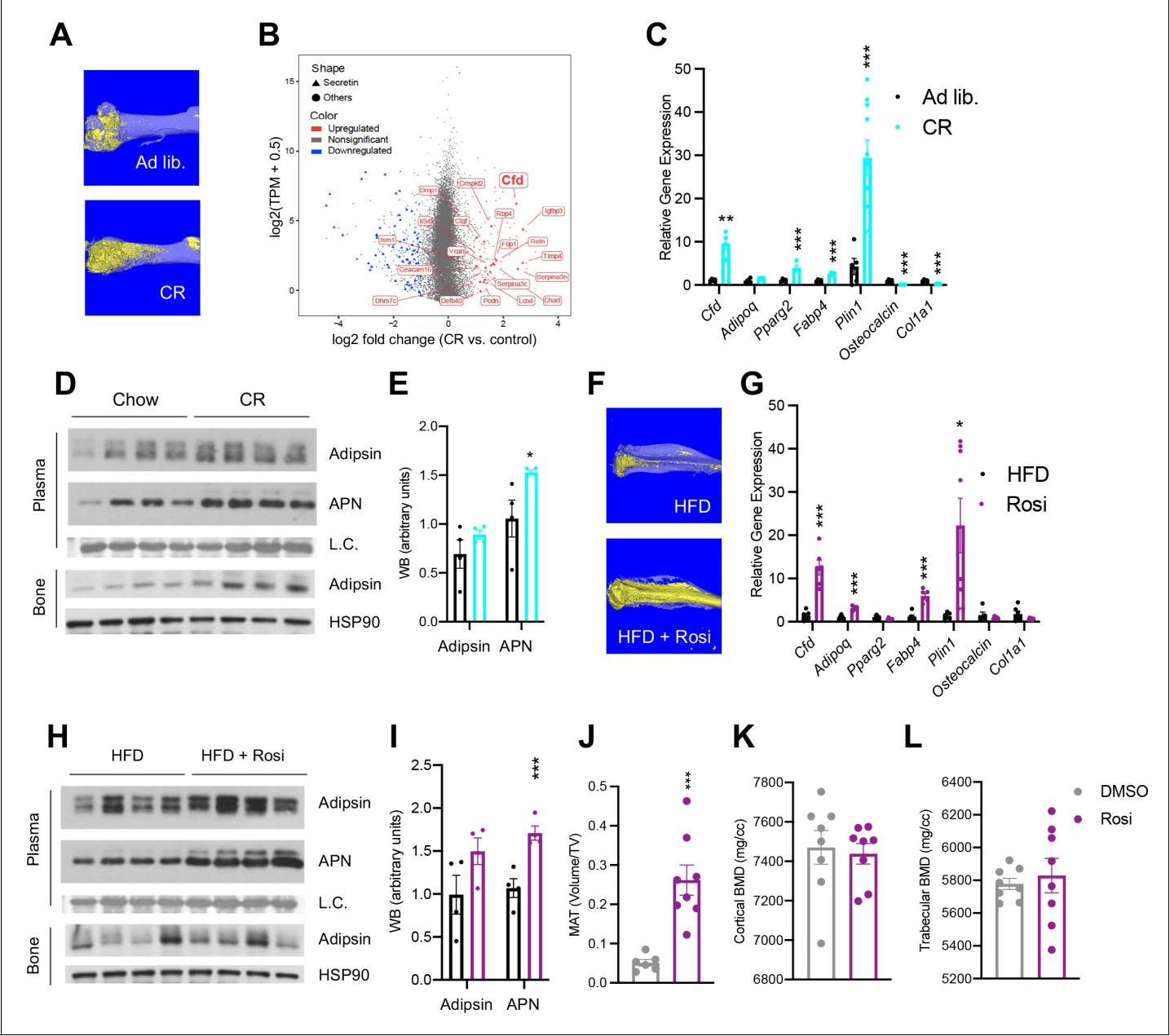

**Figure 1.** Adipsin is robustly induced in the bone marrow (BM) during bone marrow adipose tissue (BMAT) expansion. (**A**) Representative osmium tetroxide staining of MAT assessed by μCT scanning in the femurs of ad libitum and calorie restriction (CR)-fed mice. (**B**) Scatterplot displaying the gene expression (y-axis) and fold change (x-axis) in whole bone tissue between CR and control, induced secretory genes is highlighted. (**C–E**) CR-induced bone loss model: 18-week-old male mice subjected to 30% CR for 4 weeks. (**C**) qPCR analyses of gene expression in the BM isolated from the tibia (n = 6, 6); (**D**) immunoblot of plasma Adipsin and Adiponectin (APN) – Coomassie staining of the membrane was used as loading control (L.C.) (n = 4, 4); and immunoblot of Adipsin from bone – HSP90 was used as the loading control (n = 4, 4); (**E**) quantification of plasma Adipsin and APN from western blot. (**F–I**) Rosiglitazone (Rosi)-induced bone loss model: adult male mice on HFD for 12 weeks followed by 6 weeks of HFD supplemented with Rosi. (**F**) Representative osmium tetroxide staining of MAT assessed by μCT scanning in the tibia; (**G**) qPCR analyses of gene expression in the BM isolated from the femurs (n = 6, 6); (**H**) immunoblot of plasma Adipsin and APN and bone Adipsin; (**I**) quantification of plasma Adipsin and APN from western blot (n = 4, 4). (**J–L**) Adult male mice on HFD for 8 weeks followed by daily injections of 10 mg/kg Rosi or dimethyl sulfoxide solution (DMSO) (10%) for 3 weeks with continuous HFD feeding. (**J**) Quantification of femoral MAT (n = 6, 8); (**K, L**) femoral bone mineral density (BMD) in the cortical (**K**) and trabecular (**L**) regions determined by μCT scans (n = 8, 8). *p<0.05, **p<0.01, ***p<0.001 for control group vs. treatment group. Data represent mean ± SEM. Two-tailed Student's t-tests were used for statistical analyses.

The online version of this article includes the following figure supplement(s) for figure 1:

**Figure supplement 1.** The unique regulation of bone marrow (BM) Adipsin.

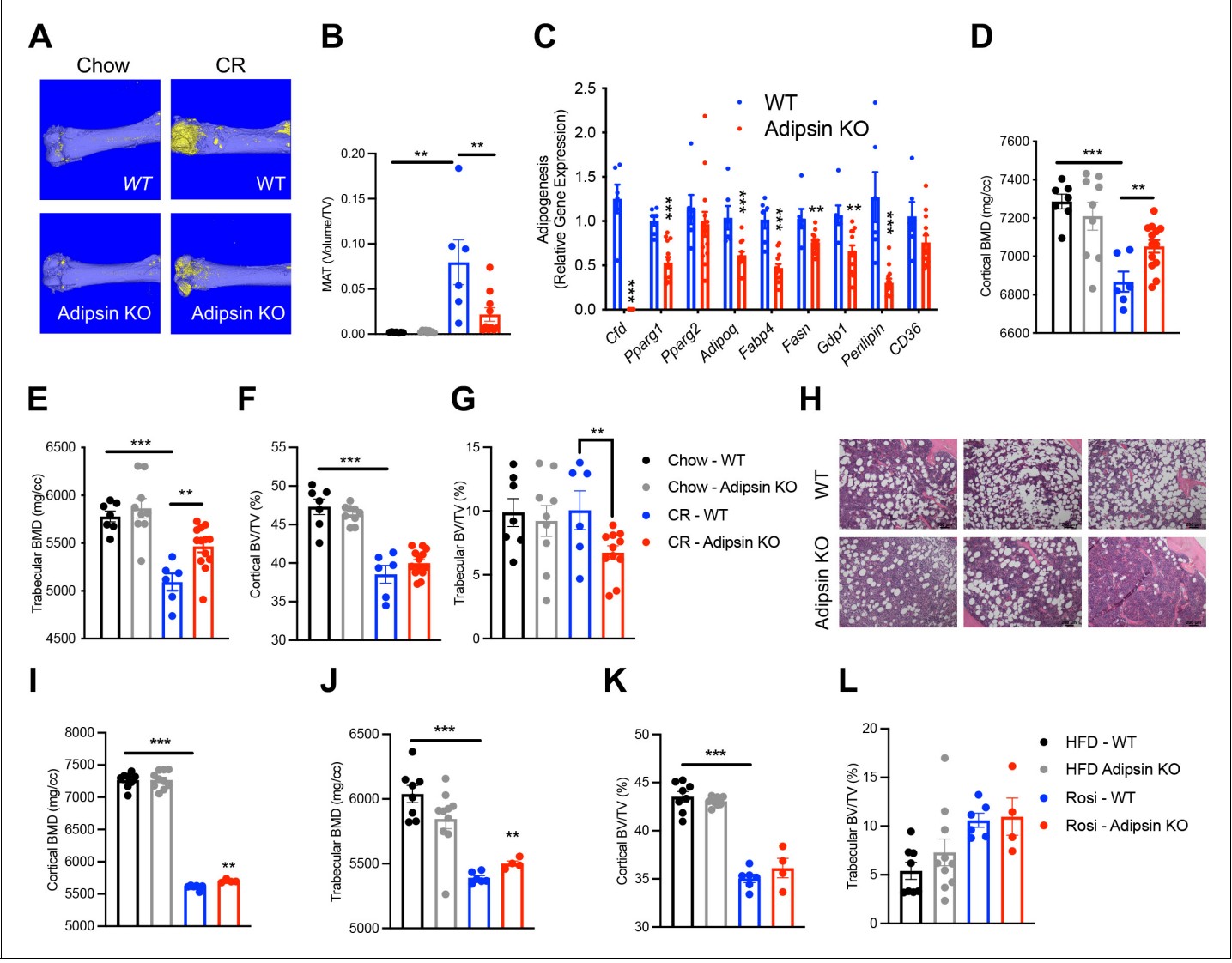

**Figure 2.** Ablation of Adipsin inhibits bone marrow (BM) adipogenesis and protects bone. (A–G) 18-week-old male mice on chow diet ad libitum or subjected to calorie restriction (CR) for 4 weeks. Chow WT (n = 7), Adipsin KO (n = 9); CR WT (n = 6), Adipsin KO (n = 10). (A) Representative osmium tetroxide staining and (B) quantification of femoral marrow adipose tissue (MAT); (C) qPCR analyses of gene expression for markers of adipocytes in the BM from the tibia of CR mice; (D, E) femoral bone mineral density (BMD) in the cortical (D) and trabecular (E) regions; (F, G) bone volume (BV) normalized by total volume (TV) in the cortical (F) and trabecular (G) regions of the femurs determined by µCT scans. (H–L) Adult male mice on HFD for 12 weeks followed by 6 weeks of HFD (n = 8, 10) or HFD supplemented with rosiglitazone (Rosi) treatment (n = 6, 4). (H) Hematoxylin and eosin stain (H&E) staining of the femoral MAT in the same region of three different mice (Rosi only); (I, J) femoral BMD in the cortical (I) and trabecular (J) regions; (K, L) BV normalized by TV in the cortical (K) and trabecular (L) regions of the femurs determined by µCT scans. *p<0.05, **p<0.01, ***p<0.001 for WT vs. Adipsin KO. Data represent mean ± SEM. Two-tailed Student's t-tests were used for statistical analyses.

The online version of this article includes the following figure supplement(s) for figure 2:

**Figure supplement 1.** Mild metabolic phenotype of Adipsin KO mice during calorie restriction (CR) and rosiglitazone (Rosi) treatment.

to CR on overall body weight, insulin sensitivity, and glucose tolerance (*Figure 2—figure supplement 1A–D*). Additionally, there were no observed differences in plasma insulin levels (*Figure 2—figure supplement 1H*). In line with their lower BMA, Adipsin KO mice showed higher BMD in the cortical and trabecular regions (*Figure 2D, E*), without significant changes between WT and Adipsin KO in trabecular number, cortical thickness (*Figure 2—figure supplement 1F, G*), or cortical bone volume (BV)/total volume (TV) but a decrease in trabecular BV/TV (*Figure 2F, G*). It should be noted that no significant changes of MAT or bone structure were observed in adult Adipsin KO mice

on ad libitum chow diet feeding (*Figure 2A–F*) and that, given these parameters, CR was successful in inducing bone loss in WT mice.

In the parallel TZD model, following the induction of obesity and insulin resistance with 12 weeks of HFD feeding, Adipsin KO and WT mice were treated with Rosi to promote MAT expansion. Adipsin KO mice showed comparable metabolic phenotypes regarding obesity and insulin sensitivity (*Figure 2—figure supplement 1I, J*), but worse glucose tolerance (*Figure 2—figure supplement 1K*) as expected (*Lo et al., 2014*; *Gómez-Banoy et al., 2019*). H&E staining revealed fewer lipid droplets accumulated in the Adipsin KO mouse BM (*Figure 2H*). Furthermore, ablation of Adipsin improved skeletal health as shown by higher BMD in both the cortical and trabecular femoral regions without affecting BV (*Figure 2I–L*), trabecular number, or cortical thickness (*Figure 2—figure supplement 1L, M*) upon Rosi treatment. As expected, these differences in the bone microarchitecture of WT and Adipsin KO mice were not observed in the HFD-only cohort, implying that Adipsin is crucial to the response and plasticity of BMAs in conditions that are known to induce BM adipogenesis. Thus, Adipsin deficiency appears to consistently alter the balance between MAT expansion and bone homeostasis in distinct bone loss models, leading to an overall bone protection effect.

## The alternative complement pathway is involved in BM homeostasis

Adipsin, also called Complement Factor D (*Rosen et al., 1989*; *White et al., 1992*), is an established activator of Complement Component 3 (C3), the central player of the complement system (*Xu et al., 2001*; *Sahu and Lambris, 2001*). C3 deficiency has been shown to protect mice from ovariectomy-induced bone loss (*MacKay et al., 2018*). Therefore, we speculated that the complement pathway could be the downstream effector through which Adipsin regulates MAT expansion. To investigate this, we challenged C3 KO mice to CR. Despite no discernible differences in body weight or composition and worse insulin sensitivity and glucose tolerance but similar fasting insulin levels (*Figure 3—figure supplement 1A–E*), C3 KO mice displayed a pronounced inhibition of MAT expansion (*Figure 3A, B*), though with minimal lipid droplets detected by H&E staining (*Figure 3—figure supplement 1F*). In contrast to MAT, the effects on bone were overall mild (*Figure 3C–F*), with increases observed only in cortical BMD (*Figure 3C*) and cortical thickness (*Figure 3—figure supplement 1G, H*). These data indicate that complement activity is critical to BM adipogenesis in response to CR.

To further establish the role of complement activation in MAT expansion, we analyzed C3 KO mice subjected to HFD to induce obesity followed by Rosi treatment. In baseline HFD-fed, obese mice C3 knockout restrained the development of MAT as shown by osmium tetroxide staining (*Figure 3G, H*) without affecting BMD (*Figure 3J, K*). This uncoupling of BMA from bone loss suggests that complement activation has a direct impact on BM adipogenesis. Strikingly, C3 deficiency prevented Rosi-induced MAT expansion by an approximately sixfold reduction (*Figure 3G, H*), further supported by observed differences in H&E staining (*Figure 3I*). Additionally, C3 KO mice displayed strong protection of cortical BMD (*Figure 3J*) and BV (BV/TV) in both the cortical and trabecular regions (*Figure 3L, M*), despite no changes in the trabecular number and cortical thickness (*Figure 3—figure supplement 1L, M*). The bone protection effect of C3 deficiency was associated with slightly improved insulin sensitivity but significantly worsened glucose tolerance (*Figure 3—figure supplement 1I–K*), in line with Adipsin KO mice (*Figure 2—figure supplement 1K*). Together, the inhibition of MAT expansion and bone loss in Adipsin KO mice is amplified in C3 KO mice, suggesting that Adipsin plays a role in regulating marrow homeostasis through complement activity.

## BM Adipsin induces MAT expansion during aging

Aging is associated with MAT expansion and decline in bone integrity (*Krings et al., 2012*). Interestingly, Adipsin in the circulation was gradually decreased during aging (*Figure 4A*), raising the question whether BM Adipsin is similarly altered with age. Indeed, *Cfd* was markedly upregulated in the BM at 78 weeks of age compared to middle-aged (26-week-old) mice (*Figure 4B*), in striking contrast to the decrease in the peripheral fat (*Figure 4C*), which potentially underlies the decrease in circulation. These data emphasize that BM Adipsin is distinct from peripheral Adipsin and is uniquely correlated with BMA and bone health during aging.

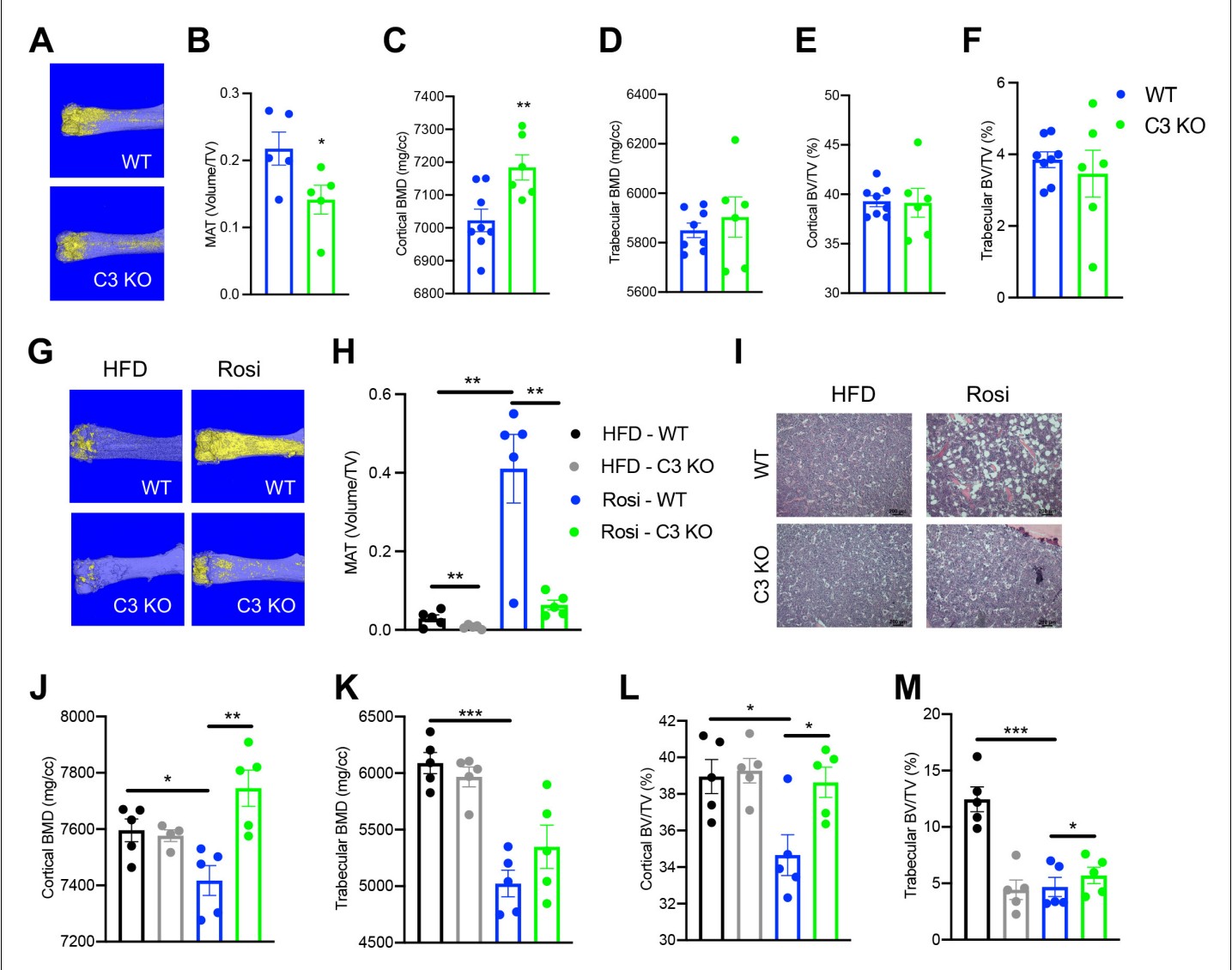

**Figure 3.** Ablation of C3 recapitulates the bone marrow phenotype of Adipsin KO mice. (A–F) 18-week-old male mice subjected to 30% calorie restriction for 4 weeks. WT (n = 8), C3 KO (n = 6). (A) Representative osmium tetroxide staining and (B) quantification of femoral marrow adipose tissue (MAT) (n = 5, 5); (C, D) femoral bone mineral density (BMD) in the cortical (C) and trabecular (D) regions; (E, F) bone volume (BV) normalized by total volume (TV) in the cortical (E) and trabecular (F) regions of the femurs determined by μCT scans. (G–M) Adult male mice on HFD for 12 weeks followed by 8 weeks of HFD or HFD supplemented with rosiglitazone (Rosi) treatment (n = 5/group). (G) Representative osmium tetroxide staining and (H) quantification of femoral MAT; (I) hematoxylin and eosin (H&E) staining of femoral MAT; (J, K) femoral BMD in the cortical (J) and trabecular (K) regions; (L, M) BV normalized by TV in the cortical (L) and trabecular (M) regions of the femurs determined by μCT scans. *p<0.05, **p<0.01 for WT vs. C3 KO. Data represent mean ± SEM. Two-tailed Student's *t*-tests were used for statistical analyses.

The online version of this article includes the following figure supplement(s) for figure 3:

**Figure supplement 1.** Metabolic phenotype of C3 KO mice during calorie restriction (CR) and rosiglitazone (Rosi) treatment.

To understand the functional significance of BM Adipsin, we assessed bone health in aging (60-week-old) Adipsin KO and WT control mice. Similar to the CR model, the osmium tetroxide staining in the femur revealed an approximately 50% decrease in BMA in Adipsin KO mice (*Figure 4D, E*), despite no discernible lipid droplet appearance in H&E staining (*Figure 4—figure supplement 1D*). Furthermore, Adipsin KO mice displayed a significantly higher BMD in the femoral trabecular region (*Figure 4G*). Though the cortical region BMD, trabecular number, and cortical thickness did not change significantly (*Figure 4F, Figure 4—figure supplement 1E, F*), the associated cortical BV/TV was increased (*Figure 4H*), indicating better bone quality overall. These bone protective effects in

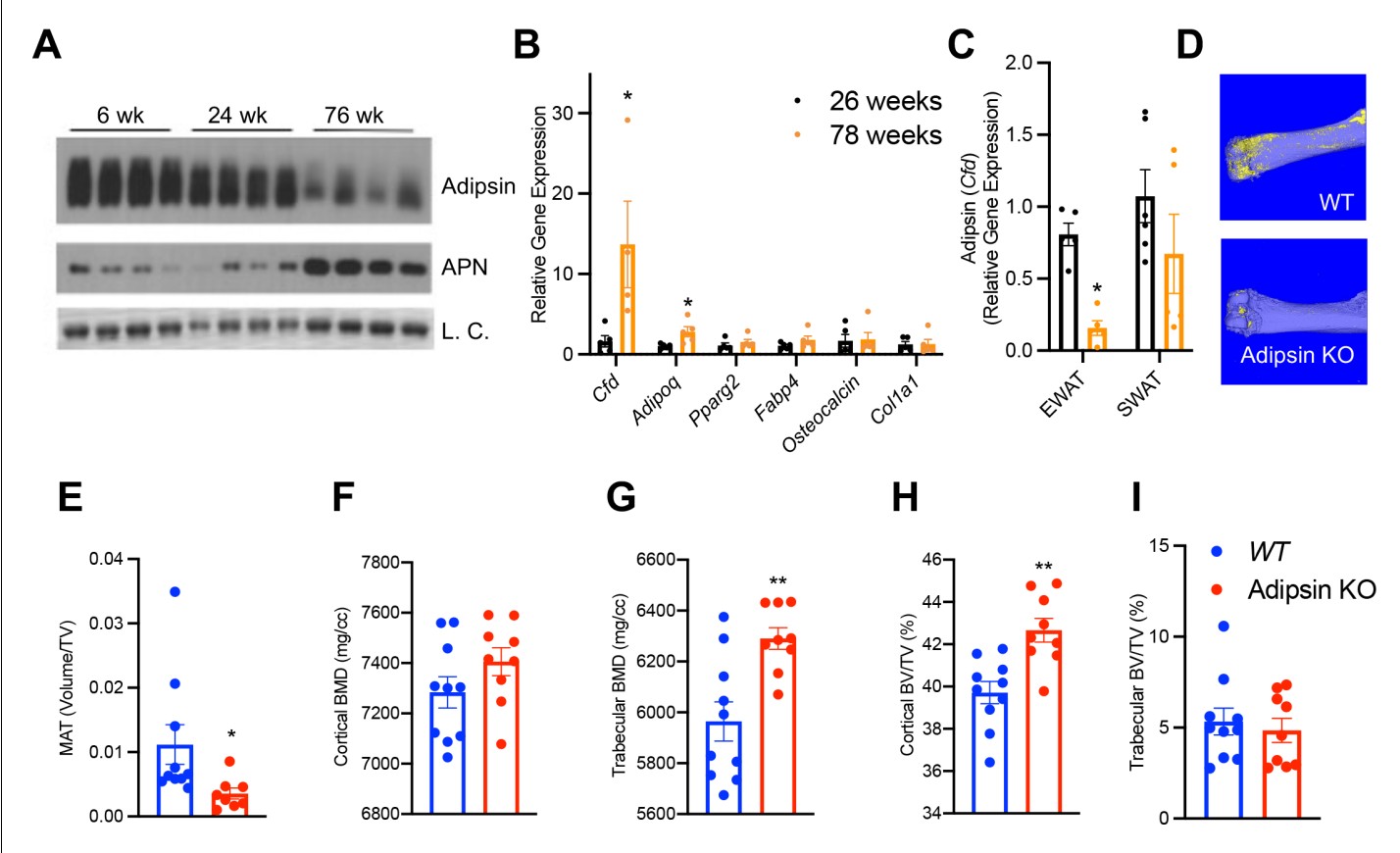

**Figure 4.** Bone marrow (BM) Adipsin induces bone marrow adiposity expansion during aging. (**A**) Immunoblot of Adipsin and Adiponectin from plasma of chow-fed male mice at 6, 24, and 76 weeks of age (L.C. = Coomassie staining of the membrane). (**B, C**) qPCR analyses of gene expression in the BM from tibia (**B**) and *Cfd* expression in the epididymal white adipose tissue and subcutaneous white adipose tissue (**C**) from chow-fed male mice at 26 and 78 weeks of age (n = 5, 5). *p<0.05 for young vs. aging mice. (**D–I**) Chow-fed 1-year-old male mice. WT (n = 10) and Adipsin KO (n = 9). (**D**) Representative osmium tetroxide staining and (**E**) quantification of femoral MAT; (**F, G**) femoral bone mineral density (BMD) in the cortical (**F**) and trabecular (**G**) regions, and (**H, I**) bone volume normalized by total voume in the cortical (**H**) and trabecular (**I**) regions of the femur determined by μCT scans. **p<0.01 for WT vs. Adipsin KO mice. Data represent mean ± SEM. Two-tailed Student's *t*-tests were used for statistical analyses.
The online version of this article includes the following figure supplement(s) for figure 4:

**Figure supplement 1.** Metabolic phenotyping of Adipsin KO mice during aging.

aging Adipsin KO mice were independent of metabolic changes as their body weight, composition, insulin sensitivity, and glucose tolerance remained comparable to those of WT (*Figure 4—figure supplement 1A–C*). These results reinforce our hypothesis that BM-derived Adipsin is favorable to BMAd development but detrimental to bone health.

## Adipsin influences the fate of BMSC differentiation

Osteoblasts and BMAds share a common progenitor, BMSCs. Given the inhibited bone loss and MAT expansion observed in Adipsin and C3 deficiency, we asked whether the differentiation of BMSCs is affected. We isolated BMSCs from WT and Adipsin KO mice and stimulated them to differentiate into adipocytes or osteoblasts. Interestingly, even with Rosi activation of PPARγ, the adipogenic capacity of Adipsin KO BMSCs was diminished, shown by decreased Oil Red O staining for lipid droplets and mRNA expression of adipocyte markers (*Figure 5A, B*). On the other hand, these cells were more readily differentiated into osteoblasts as determined by increased Alizarin Red staining for mineralization and increased expression of osteoblast markers, including *Runx2, Atf4, Osx, Alpl, Ptfhr, Col1a1*, and *Osteocalcin* (*Figure 5C, D*). To assess whether this impaired adipogenesis was a generalized feature of Adipsin deficiency in adipocyte precursor cells, we isolated the ASCs from the SWAT of WT and Adipsin KO mice and stimulated their differentiation into adipocytes.

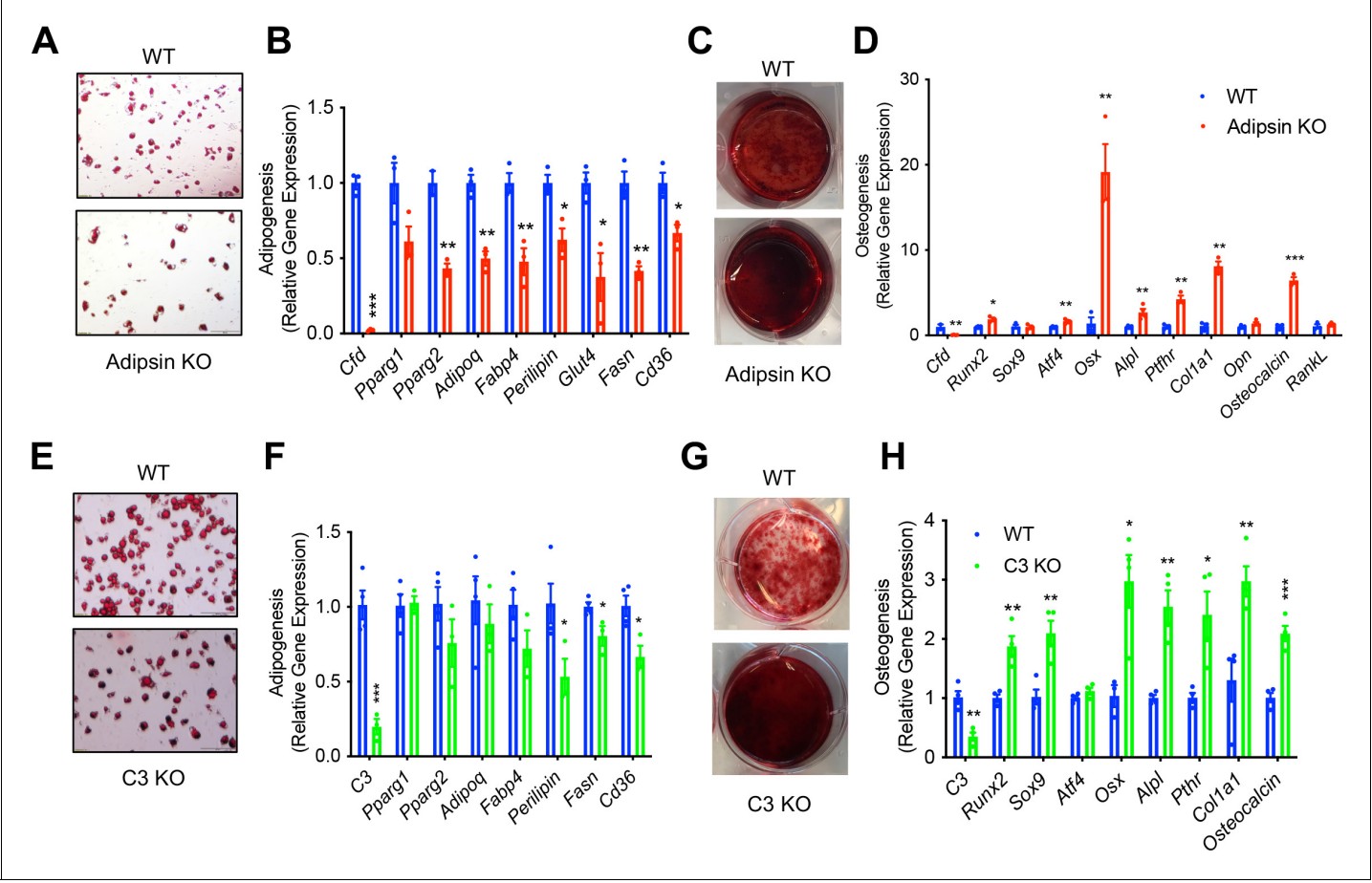

**Figure 5.** Adipsin influences the fate of bone marrow stromal cell (BMSC) differentiation. (A, B) Adipogenesis of WT and Adipsin KO BMSCs. (A) Oil Red O staining of lipid droplets after 2 weeks of differentiation; (B) qPCR analysis of adipogenic genes (n = 4, 4). (C, D) Osteoblastogenesis of WT and Adipsin KO BMSCs. (C) Alizarin Red staining of calcium after 21 days of differentiation; (D) qPCR analysis of osteoblastogenic genes (n = 4, 4). (E, F) Adipogenesis of WT and C3 KO BMSCs. (E) Oil Red O staining of lipid droplets after 2 weeks of differentiation; (F) qPCR analysis of adipogenic genes (n = 4, 4). (G, H) Osteoblastogenesis of WT and C3 KO BMSCs. (G) Alizarin Red staining of calcium after 21 days of differentiation; (H) qPCR analysis of osteoblastogenic genes (n = 4, 4). *p<0.05, **p<0.01, ***p<0.001 for WT vs. mutant cells. Data represent mean ± SEM. Two-tailed Student's *t*-tests were used for statistical analyses.

The online version of this article includes the following figure supplement(s) for figure 5:

**Figure supplement 1.** Absence of an effect of Adipsin in subcutaneous white adipose tissue (SWAT)-derived adipose stromal cells.

However, the adipogenic capacity for the ASCs from Adipsin KO mice was not hindered indicated by comparable Oil Red O staining and adipogenic gene expression (*Figure 5—figure supplement 1A, B*). The difference in differentiation capability between BMSCs and ASCs highlights the specific effect of Adipsin on the BM.

Given the overlapping effects of Adipsin and C3 deficiencies on BM homeostasis in vivo, we further suspected that Adipsin drives changes in BM through its activation of C3 in the alternative complement pathway. Similar to Adipsin KO cells, C3 KO BMSCs had impaired adipogenesis, as shown by lower Oil Red O staining and modest downregulation of markers related to adipogenesis (*Figure 5E, F*). Conversely, the ablation of C3 enhanced osteoblast differentiation capacity, indicated by increased Alizarin Red staining and upregulation of mRNA expression of osteoblastogenesis markers (*Figure 5G, H*). Together, these results suggest that Adipsin deficiency directly modulates BMSC fate by hindering their differentiation into adipocytes and favoring osteoblastogenesis through complement activity.

## Adipsin primes BMSCs toward adipogenesis through inhibition of Wnt signaling

Due to the previously observed influence of Adipsin on the overall BM microenvironment, we sought to determine whether Adipsin KO BMSCs were primed toward osteoblastogenesis prior to induction. Confluent, undifferentiated Adipsin KO and WT BMSCs were treated with 1 µM dexamethasone, a glucocorticoid commonly used in vitro to stimulate both adipogenesis and osteoblastogenesis (*Chen et al., 2016*). We reasoned that the short-term dexamethasone treatment would initiate differentiation but not fully determine lineage preference. Interestingly, at 48 hr post-treatment, Adipsin KO cells displayed a higher expression of genes associated with osteoblastogenesis, including *Runx2*, *Atf4*, and *Opn*, whereas adipocyte markers *Adipoq*, *Fabp4*, *Fasn*, and *Cd36* were significantly repressed (*Figure 6A*). These data suggest that Adipsin KO BMSCs may be primed toward osteoblastogenic differentiation. As nascent wild-type BMSCs barely express adipogenic genes such as Adipsin, these results further imply that the presence of Adipsin in the BM microenvironment primes BMSC fate toward adipogenesis in vivo.

To obtain direct evidence of the priming of BMSCs by Adipsin, we treated BMSCs isolated from Adipsin KO mice, which are naïve to Adipsin exposure, with recombinant Adipsin. Adipsin treatment repressed the induction of osteoblast markers during osteoblastogenic differentiation (*Figure 6B*). On the other hand, exogenous Adipsin had a mild effect on promoting adipocyte differentiation, as shown by increased lipid accumulation and upregulation of PPARγ2, the master regulator of adipogenesis (*Figure 6—figure supplement 1A–C*). Conversely, inhibiting the downstream C3 complement activity by SB290157, a C3aR inhibitor, prevented lipid accumulation and repressed adipogenic gene expression during adipogenesis (*Figure 6C, Figure 6—figure supplement 1D, E*) while promoting the upregulation of osteoblast transcription factor *Osx* during osteoblastogenesis (*Figure 6—figure supplement 1F*) in BMSCs. Moreover, treating BMSCs with SB290157 blunted the effects of exogenous Adipsin on their differentiation into adipocytes and osteoblasts (*Figure 6—figure supplement 1G, H*). Together, these data suggest that complement activity suppression is optimal for osteoblast differentiation and inhibitory of adipogenesis. These data support our conclusion that Adipsin functions in the BM microenvironment to prime BMSCs toward adipocyte differentiation and inhibit osteoblastogenesis.

To understand the mechanism underlying the priming of BMSCs, we assessed whether intrinsic transcriptional differences already existed in undifferentiated Adipsin KO BMSCs. We found a notable upregulation in genes associated with the canonical (*Wwtr1*, *Lrp5*) and non-canonical Wnt pathways (*Wnt5a*) (*Figure 6D*). In the canonical Wnt pathway, Wnt ligand inactivates GSK3 via phosphorylation to inhibit the phosphorylation of β-catenin, leading the latter to stabilize and translocate into the nucleus in order to transcribe downstream target genes (*Wu and Pan, 2010*). This pathway is well established to inhibit adipogenesis (*Park et al., 2019*; *Kim et al., 2010*) and promote bone formation (*Gong et al., 2001*; *Mani et al., 2007*; *Glass and Karsenty, 2007*); while upregulation of non-canonical-associated proteins induces Runx2 expression (*Takada et al., 2007*; *Tu et al., 2007*), the main osteoblastogenic transcriptional factor. Thus, Adipsin likely inhibits Wnt signaling in order to determine the differentiation fate of BMSCs. Consistently, Adipsin treatment inhibited both the canonical (*Lrp5*) and non-canonical (*Wnt5a*) Wnt signaling genes, whereas SB290157 treatment abolished this inhibition in Adipsin KO BMSCs (*Figure 6E*). In support of this hypothesis, Adipsin treatment blunted Wnt3a-induced phosphorylation of GSK3β and prevented the subsequent accumulation of β-catenin in C3H10T1/2 mesenchymal stem cells (*Figure 6F*). Furthermore, the activation of canonical Wnt signaling was recapitulated in Adipsin KO mice, which displayed increased GSK3β phosphorylation in the bone (*Figure 6G*). Though β-catenin signal was too low to be detected by western blotting, immunohistochemical staining revealed an increase in β-catenin in the BM of Adipsin KO mice (*Figure 6H*). Additionally, Wnt/β-catenin signaling pathway genes *Wnt3a*, *Fzd9*, and *Ctnnb1* (*Rudloff and Kemler, 2012*), but not the β-catenin-repressive *Tcf3* and *Tcf7l2*, were upregulated in the BM of Adipsin KO mice after Rosi treatment (*Figure 6I*). There were no significant changes of p-GSK3β and β-catenin observed in the EWAT (*Figure 6—figure supplement 2A*), emphasizing a bone-specific role for Adipsin. We also observed increases in GSK3β phosphorylation, β-catenin, and expression of Wnt pathway genes in the bone of C3 KO mice after Rosi treatment (*Figure 6J–L*), despite no discernible differences in the EWAT (*Figure 6—figure supplement 2B*). The Adipsin KO and C3 KO mice on CR recapitulated these changes to Wnt signaling

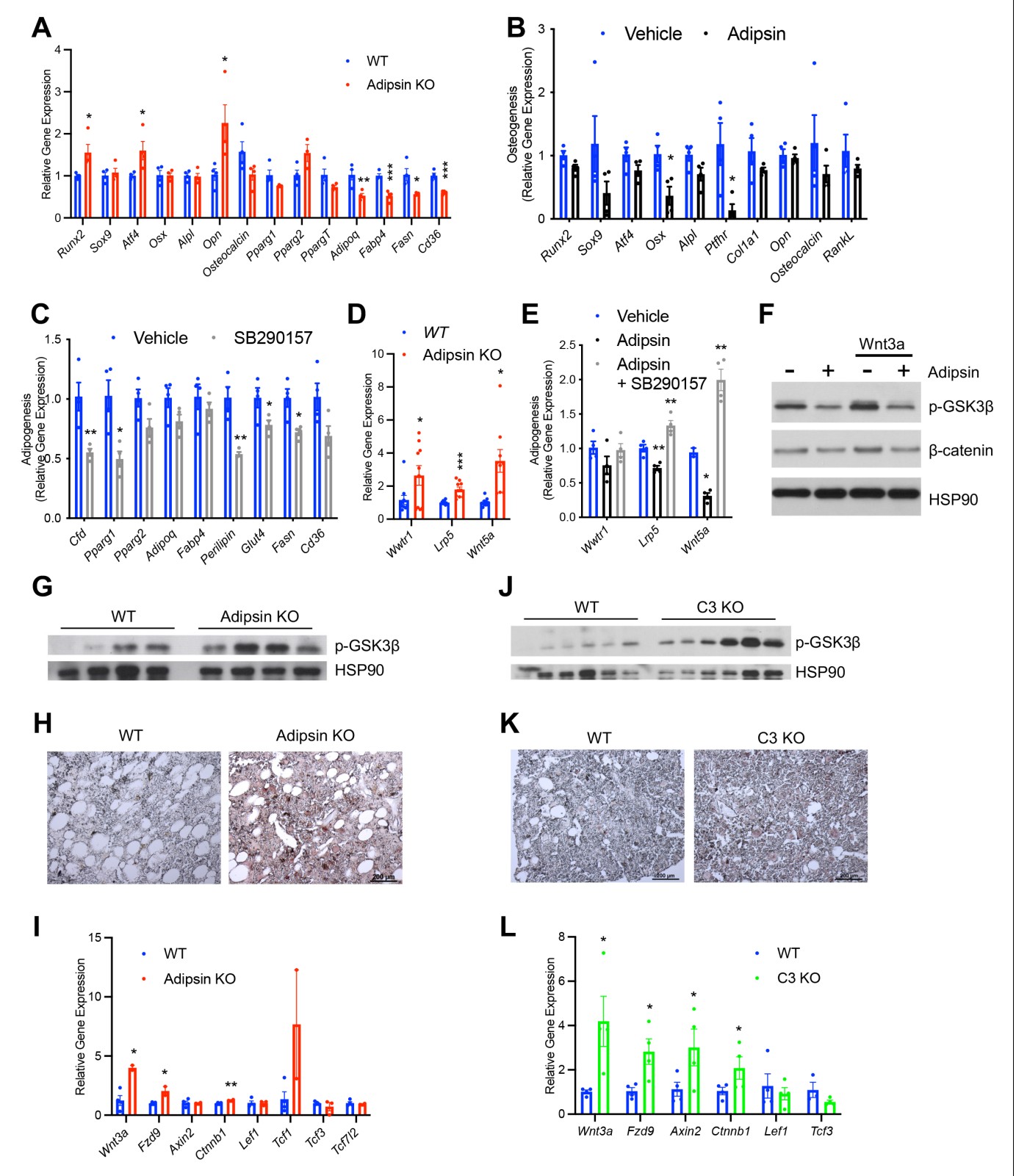

**Figure 6.** Adipsin primes bone marrow stromal cells (BMSCs) toward adipogenesis through inhibition of Wnt signaling. (**A**) qPCR analysis of adipogenic and osteoblastogenic genes in BMSCs isolated from WT and Adipsin KO mice prior to differentiation following 48 hr of dexamethasone (1 μM) treatment. *p<0.05, **p<0.01, ***p<0.001 for WT vs. Adipsin KO BMSCs (n = 4, 4). (**B**) qPCR analysis of osteoblastogenic markers in Adipsin KO BMSCs differentiated into osteoblasts with or without recombinant mouse Adipsin (1 μg/mL) treatment. *p<0.05 for vehicle vs. Adipsin (n = 4, 4). (**C**) qPCR

*Figure 6 continued on next page*

Figure 6 continued

analysis of adipogenic markers in WT BMSCs differentiated into adipocytes with or without C3aR antagonist SB290157 (1 µM) treatment. *p<0.05, **p<0.01 for vehicle vs. SB290157 (n = 4, 4). (D) qPCR analysis of genes associated with Wnt pathway activation in undifferentiated BMSCs isolated from WT and Adipsin KO mice. *p<0.05, ***p<0.001 for WT vs. Adipsin KO BMSCs (n = 4, 4). (E) qPCR analysis of genes associated with Wnt pathway activation in Adipsin KO BMSCs treated with recombinant Adipsin (1 µg/mL) or SB290157 (1 µM). *p<0.05, **p<0.01, ***p<0.001 for vehicle vs. treatment (n = 4/group). (F) Immunoblot of phospho-GSK3β and β-catenin from C3H10T1/2 cells treated with Adipsin (1 µg/mL) and Wnt3a (20 ng/mL) (L.C. = HSP90). (G–I) Immunoblot of phospho-GSK3β (L.C. = HSP90) (G), immunohistochemical staining of β-catenin (H), and qPCR analysis of Wnt signaling markers (I) in the femurs of WT and Adipsin KO mice on rosiglitazone (Rosi) diet (n = 4, 4, bones lost due to harvesting and processing). (J–L) Immunoblot of phospho-GSK3β (L.C. = HSP90) (J), immunohistochemical staining of β-catenin (K), and qPCR analysis of Wnt signaling markers (L) in the femurs of WT and C3 KO mice on Rosi diet (n = 4, 4, bones lost due to harvesting and processing). *p<0.05, **p<0.01 for WT vs. mutant. Data represent mean ± SEM. Two-tailed Student's t-tests were used for statistical analyses.

The online version of this article includes the following figure supplement(s) for figure 6:

**Figure supplement 1.** Adipsin modulates bone marrow stromal cell (BMSC) differentiation and Wnt signaling *in vitro*.
**Figure supplement 2.** Adipsin modulates bone marrow stromal cell (BMSC) differentiation and Wnt signaling *in vitro*.

markers as well (*Figure 6—figure supplement 2C-F*). Together, these data imply that Adipsin inhibits Wnt signaling through its complement activity to prime BMSCs toward adipocyte differentiation.

## Adipsin is a downstream target of PPARγ deacetylation

Next, we sought to understand how Adipsin is regulated. Adipsin is induced during adipogenesis by PPARγ, the master regulator of adipocyte biology (*Tontonoz et al., 1994*). Adipocyte-specific ablation of PPARγ leads to significantly increased trabecular bone density with an associated absence of BMA accompanied by a lipoatrophy phenotype (*Wang et al., 2013*). Notably, these knockouts displayed a complete absence of circulating Adipsin (*Figure 7A*), suggesting that PPARγ is essential for Adipsin production. Furthermore, we have observed that *Cfd* expression is sensitive to PPARγ acetylation changes in our previous studies (*Qiang et al., 2012*; *Kraakman et al., 2018*). We, therefore, asked whether Adipsin is a downstream target of PPARγ deacetylation. As predicted, Adipsin levels in the EWAT and in circulation were markedly decreased in the PPARγ deacetylation-mimetic 2KR mice, while other adipokines and adipocyte markers, such as Adiponectin and aP2, were not as significantly altered (*Figure 7B*). This decrease of circulating Adipsin was supported by prevalent repression of *Cfd* in three major adipose depots – EWAT, SWAT, and brown adipose tissue (BAT) (*Figure 7—figure supplement 1A–C*) as well as in the BM of 2KR mice (*Figure 7—figure supplement 1D*). This is in contrast to the largely normal expression of other adipocyte genes (*Kraakman et al., 2018*). Of note, the 2KR model largely recapitulates the bone protection and inhibited MAT expansion effect observed in adipocyte conditional PPARγ KO mice but without the previously observed lipoatrophy (*Kraakman et al., 2018*). The repression of Adipsin in these two PPARγ mutant mouse models is consistent with the inhibited BMA and improved skeletal health observed in Adipsin KO and C3 KO models.

We then investigated the mechanism whereby repression of Adipsin by PPARγ deacetylation occurs. In HEK293T cells, Adipsin promoter was activated >300-fold by WT PPARγ in a luciferase reporter assay, while the 2KR mutant showed a much weaker transcriptional activity, especially in response to activation by Rosi treatment (*Figure 7C*), recapitulating the in vivo results (*Figure 7B*) and indicating a direct regulation of PPARγ on the Adipsin promoter. We further showed that WT PPARγ directly bound to the Adipsin promoter while the *2KR* mutation impaired this binding at multiple regions, particularly in the regions of −448 to −355 bp, −252 to −110 bp, and −114 to +33 bp identified by ChIP in comparably differentiated adipocytes (*Figure 7D*). Further manipulation of the promoter by truncated deletions revealed that the −281 to +35 bp region accounted for major promoter activation by PPARγ, with a sequence at −165 to −50 bp necessary for maximal promoter activity (*Figure 7E–G*). Thus, PPARγ deacetylation on Lys268 and Lys293, though localized to the ligand-binding domain (LBD), impairs the recruitment of PPARγ to the Adipsin promoter, underlying the repressed expression. Ultimately, the distinct regulation of Adipsin by PPARγ acetylation as well as the role of Adipsin as a secretory protein that functions on BM progenitor cells provides added emphasis for Adipsin as a mediator of BM homeostasis.

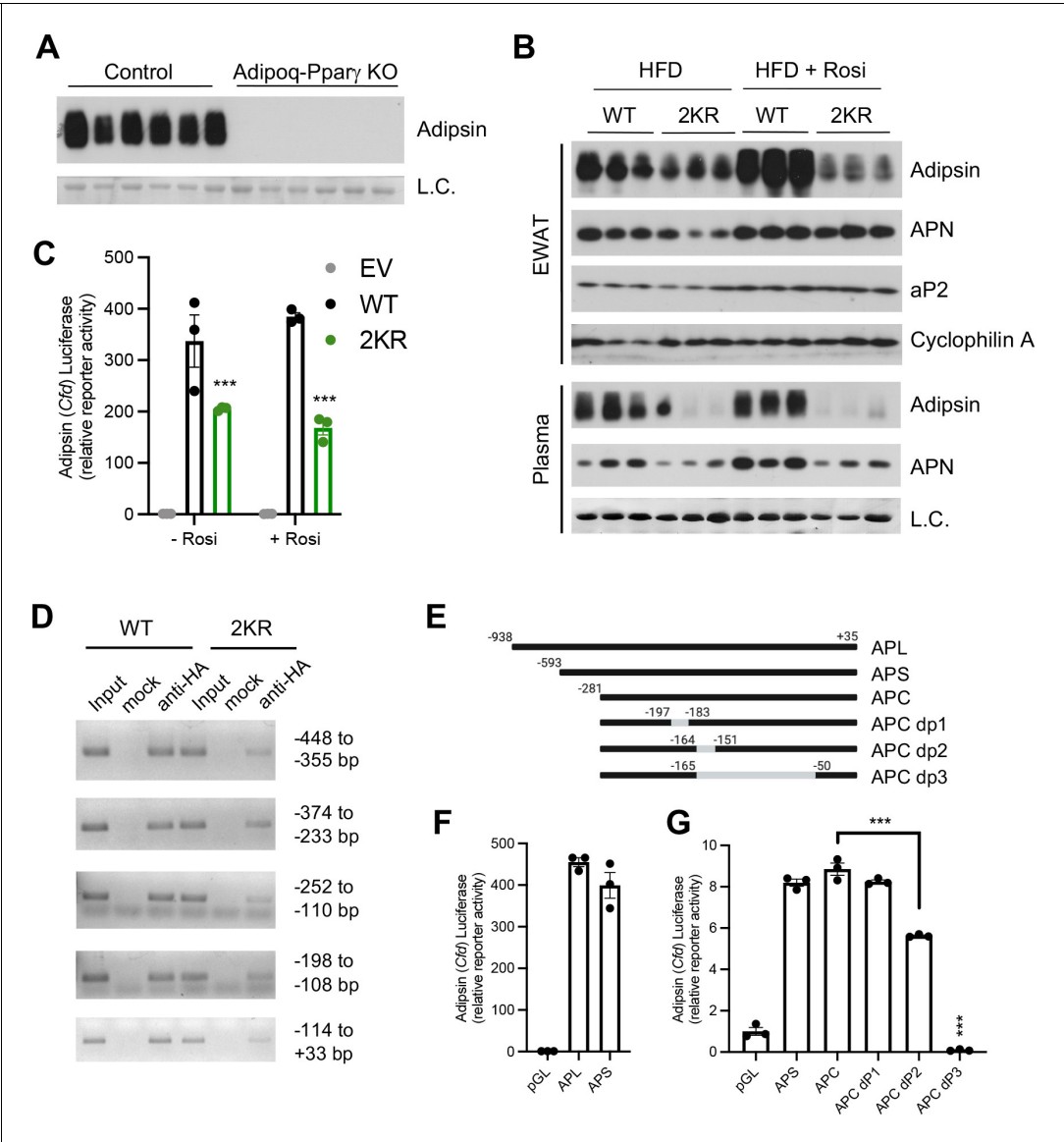

**Figure 7.** Adipsin is a downstream target of PPARγ deacetylation. (**A**) Immunoblot of Adipsin in the plasma from adult male adipocyte conditional PPARγ KO (Adipoq-Pparγ KO) and control mice on HFD for 12 weeks (L.C. = Coomassie staining of the membrane). (**B**) Adult male WT and 2KR mice on HFD for 12 weeks followed by 8 weeks of HFD or HFD supplemented with rosiglitazone (Rosi). Immunoblots of Adipsin, Adiponectin (APN), and aP2 from epididymal white adipose tissue (EWAT) (loading control = cyclophilin A) and Adipsin and APN from plasma (L.C. = Coomassie stain of the membrane). (**C**) Adipsin promoter-driven luciferase reporter assay from HEK293T cells transfected with WT or 2KR overexpression of PPARγ with or without Rosi treatment (n = 3/group). (**D**) ChIP assay for PPARγ binding to the Adipsin promoter. *Pparg*⁻/⁻ mouse embryonic fibroblasts (MEFs) were reconstituted with Flag-HA-tagged WT or 2KR PPARγ2 and adipogenesis was induced. Anti-HA ChIP assay was performed on day 7 of differentiation. (**E**) Scheme of Adipsin promoter designs: long Adipsin promoter (APL), short Adipsin promoter (APS), Adipsin promoter core (APC), delete −197 to −183 (dP1), delete −164 to −151 (dP2), delete −165 to −50 (dP3). (**F, G**) Adipsin promoter-driven luciferase reporter assay in HEK293T cells with various deletions in the Adipsin promoter region. ***p<0.001 for WT vs. 2KR. Data represent mean ± SEM. Two-tailed Student's *t*-tests were used for statistical analyses.

The online version of this article includes the following figure supplement(s) for figure 7:

**Figure supplement 1.** PPARγ deacetylation represses Adipsin expression in peripheral adipose tissues.

## Adipsin is upregulated in humans with MAT expansion

In human, fasting is a potent signal that can induce an increase in MAT. To understand the paradox that both fasting and high-calorie diet drive marrow adipogenesis, we examined gene expression patterns in BMAds isolated from BM aspirates in a previous study of high calorie and fasting in

human volunteers (*Fazeli et al., 2021*). 11 volunteers (six females, five males) were fasted for 10 days and had pre- and post-fast marrow aspirate analyzed by MR spectroscopy and bone resorption markers among other phenotypic studies. By spectroscopy, vertebral MAT increased 8.1 ± 2.6% (p=0.01) at day 10 and bone resorption increased by 77% (p<0.0001). 5 (three females, two males) of the 11 individual marrow samples (with adequate both pre- and post-samples) had qualitative BMAd RNA that met criteria for subsequent RNA-seq analysis. The top 250 genes were selected based on the highest and lowest fold change and smallest p-value. Complement activation was the second most upregulated pathway from pre- to post-10 day fast (*Figure 8A*). Pairwise analysis for the five samples between pre- and post-fasted mRNA revealed that Adipsin was increased 2.8-fold (p=$9.5 \times 10^{-9}$) after fasting, in line with the fifth upregulated pathway of upstream PPAR signaling. 229 protein-coding genes were mapped by STRING with CFD at the center region (*Figure 8B*), suggesting an active role of Adipsin in the regulatory network. Additionally, heat maps were generated using GO terms for complement activation including 173 genes. 32 genes met the cutoff criteria, including Adipsin. The complement pathway genes were markedly upregulated, particularly CFD (Adipsin), which clustered with other complement-related genes (*Figure 8C*).

## Discussion

In the present study, we demonstrated a positive correlation between Adipsin and MAT expansion and, conversely, an inverse relationship between Adipsin and skeletal integrity in multiple bone-altering mouse models, including CR, TZD, aging, 2KR, and adipocyte-specific PPARγ KO. A similar finding was recapitulated in human BMAds after a 10-day fast. We further employed Adipsin KO and C3 KO mice to demonstrate a complement-dependence tilting of BM homeostasis toward adipogenesis and away from osteoblastogenesis. Adipsin appears to execute this function through inhibition of Wnt signaling and, thus, primes BMSCs toward adipocyte differentiation (*Figure 9*). Our work reveals Adipsin to be an important regulator of BM homeostasis, serving as a previously unknown link between adipose tissue and bone.

Adipokines are thought to perform diverse roles pertaining to the endocrine regulation of skeletal remodeling. Leptin, for example, is considered beneficial for bone health as its deficiency is associated with reduced bone growth (*Turner et al., 2013*) while its restoration normalizes bone density (*Upadhyay et al., 2015*). On the other hand, Adiponectin has been shown to hinder osteoblast proliferation and promote apoptosis (*Kajimura et al., 2013*). To our surprise, the role of Adipsin in the BM is often overlooked despite its abundant production by adipocytes. It is well understood that MAT develops in pathophysiological conditions or under metabolic stress. We have shown that more Adipsin is consequently produced in the BM to promote further BMA expansion at the expense of bone formation. In this Adipsin-mediated manner, BM can quickly adapt to environmental changes. Thus, we propose a mechanism to sustain BM plasticity through Adipsin signaling. For example, in the CR model of Adipsin KO, peripheral fat mass was higher in contrast to the lower MAT, possibly owing to inefficient lipid mobilization to the BM in the absence of Adipsin. Our findings will help to understand the development of MAT and the reciprocal relationship between bone health and MAT, which are not sufficiently addressed by previously studied adipokines.

The production of Adipsin by peripheral fat has been long studied. However, no previous study has highlighted the potential role of the BM as an important source of Adipsin. The marrow provides a particularly unique microenvironment that is influenced by BMAd biology. MAT is a distinct fat depot recognized for its unique localization, regulation, and function (*Hardouin et al., 2016*). Our in vitro model has proved that without Adipsin, BMSCs have an attenuated capacity for adipogenesis even after PPARγ activation by Rosi, highlighting a direct effect of Adipsin on the BM niche in vivo. Furthermore, in conditions of low Adipsin expression such as in progenitor and osteoblast cells, the changes observed in the Adipsin KO models implicate the potential paracrine regulation of Adipsin in vivo that consequently affects the primary BMSCs. In contrast, the absence of an effect of Adipsin ablation in ASCs derived from peripheral fat is likely due to the predetermined fate of these adipocyte progenitor cells as well as the unique environment that may affect Wnt signaling.

Though the production of Adipsin in the BM is relatively low compared to other adipose tissues, its local concentration is not negligible. Of note, we observed that the inhibition of Adipsin has no direct impact on cell number or cellular composition at the basal level, ultimately highlighting the role of Adipsin as a responsive target to changes in the BM microenvironment. Thus, the role of

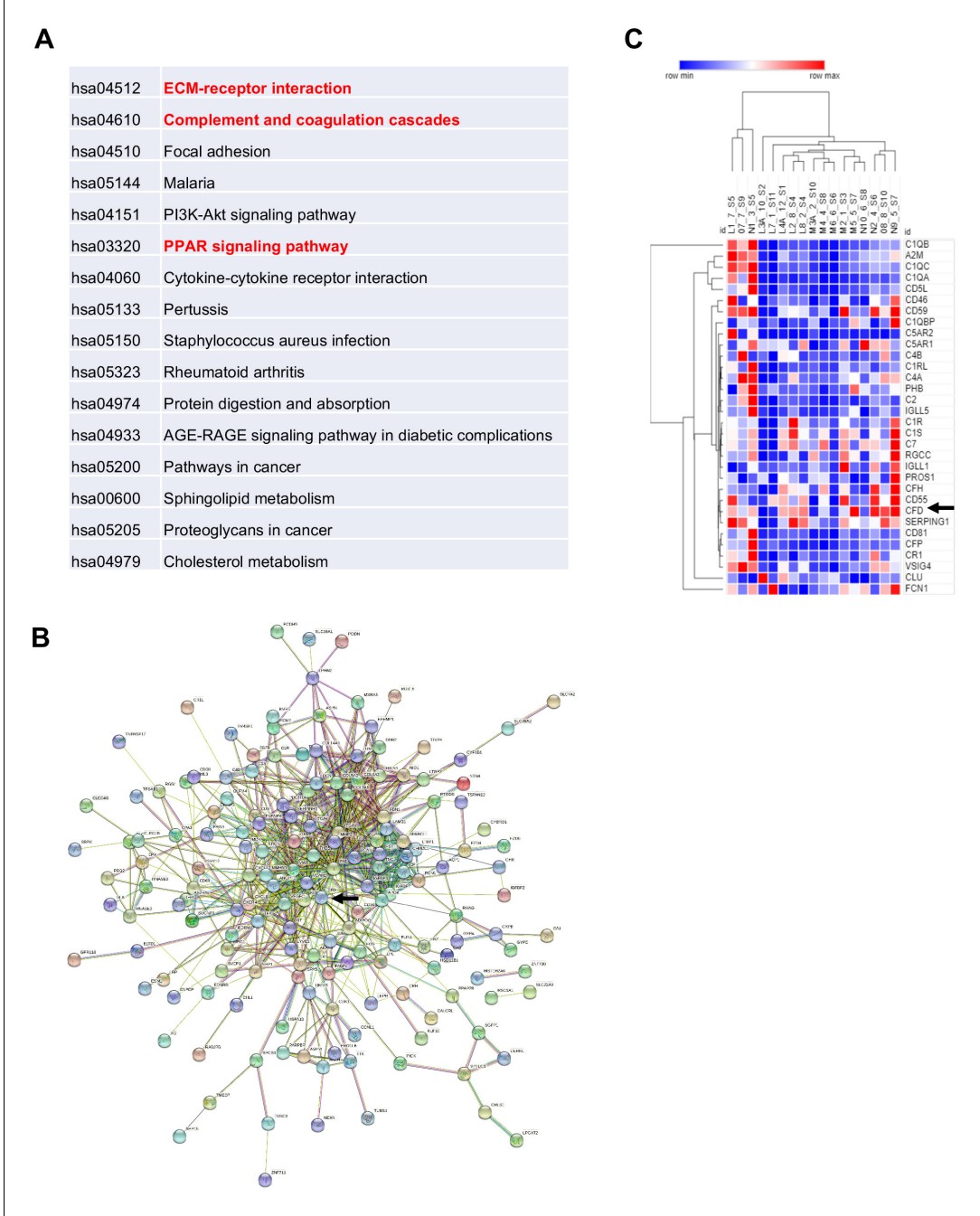

**Figure 8.** Adipsin is induced in human bone marrow adipose tissue during fasting. Five subjects (three females and two males) had qualified RNA from paired bone marrow adipocyte samples for RNA-seq. Analysis was performed by STRING and protein-coding genes were assessed with p-value<0.05, false discovery rate < 0.05. (A) Enriched pathways of selected genes; (B) STRING map, with Complement Factor D (CFD) (Adipsin) indicated by arrow; (C) heat map of the complement pathway. Five individuals are denoted by letters L–N and their respective number for pre- and post-sample time. Arrows represent CFD (Adipsin).

Adipsin in the BM becomes particularly important in conditions of MAT expansion when Adipsin is robustly induced. Furthermore, the dissociation of BM Adipsin from circulating levels in aging mice reinforces a paracrine function of Adipsin in the BM niche. Our findings help provide an explanation of a previous observation that changes in peripheral adipose tissues have minimal impact on osteogenesis (*Zou et al., 2020*). Future investigations should note the distinction between Adipsin sourced from the BM and peripheral adipose tissue. This could be achieved by developing an *Cfd*

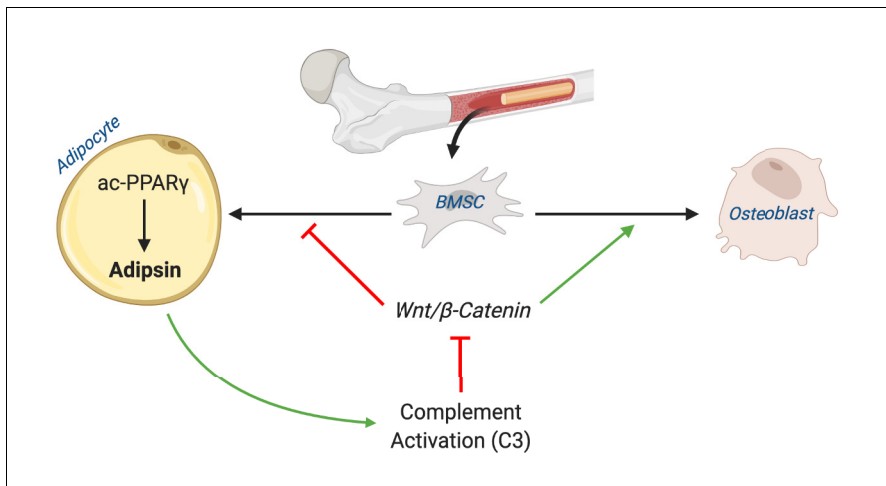

**Figure 9.** Schematic model. Adipsin from bone marrow adipocytes regulates bone marrow stromal cell fate determination through its activation of the complement system (C3) and, ultimately, downstream inhibition of the canonical Wnt signaling cascade through β-catenin. Furthermore, Adipsin transcription is determined by the acetylation post-translational modification of PPARγ.

floxed mouse model to specifically ablate Adipsin from the BM to examine the tissue-specific role of Adipsin.

The complement system has been identified as an important regulator of bone turnover and development. Previous studies have shown that the production of C3 by osteoblasts induces the differentiation of BM cells into osteoclasts (*Hardouin et al., 2016*; *Roche-Molina et al., 2015*; *Szulc et al., 2017*). Furthermore, male mice lacking CD59, a negative regulator of the complement system, show lower BMD (*Bloom et al., 2016*). These compelling data offer strong support to our premise that Adipsin may exert its effect through alternative complement activity. Furthermore, this potential mechanism is strengthened by our data showing that the C3 KO mice displayed consistent phenotypes, as in Adipsin KO mice, on inhibiting BMA and improving bone mass in response to CR and TZD. It should be noted that the BMA-inhibitory effect of C3 KO is considerably stronger than Adipsin KO. Additionally, the effect of C3 inhibition is stronger on MAT than on the bone microarchitecture in vivo. These data reflect the complexity of C3 as it is also involved in the lectin and classical complement pathways in parallel with the Adipsin-induced alternative pathway in the complement system. Further dissection of the role of Adipsin in the bone would include the inhibition of the two other complement pathways, such as by manipulating C2 or C4, which are critical complement components but not involved in the alternative pathway.

The regulation of Adipsin appears to be unique and intriguing. Its circulating and expression levels are severely impaired in genetic obesity *ob/ob* and *db/db* mice and decreased after prolonged fat expansion (*Rosen et al., 1989*). Adipsin is an adipocyte-specific gene that requires PPARγ activation (*Tontonoz et al., 1994*). We have observed that circulating Adipsin in WT mice increased by Rosi (*Figure 7B*) but have also observed that Adipsin responds in a repressive manner by PPARγ activation using TZDs in vitro. Furthermore, Adipsin is sensitive to PPARγ PTM as the deacetylation-mimetic in 2KR mice display striking and specific repression of Adipsin. We showed here that 2KR exerted repression on Adipsin by impeding its binding to the Adipsin promoter even though the modified residues both localize to the LBD, not the DNA-binding domain (DBD). Thereby, it is plausible to speculate that PPARγ deacetylation affects co-factors interacting with the DBD and, thus, impairs its affinity for DNA-binding. For example, we have also shown that deacetylated PPARγ preferentially interacts with PRDM16 and disrupts the binding of the transcriptional corepressor NCoR in regulating brown adipocyte gene *Ucp1*'s expression (*Qiang et al., 2012*). Our findings, proving the interaction between PPARγ and the Adipsin promoter, support a body of evidence that the acetylation dynamics of PPARγ regulates its pleiotropic functions and highlights its role in the BM niche.

Though its potent insulin-sensitizing effects are well documented, the widespread use of TZDs has drastically declined because of its side effects, including increased bone fragility. Our finding of

Adipsin as a priming factor of BMSCs helps to explain why 2KR mice have protected bone loss and reduced BMA phenotypes with TZD treatment. In this regard, the novel function of Adipsin may provide a new avenue for treating bone loss in diabetic patients, especially as a co-therapy for those receiving TZD treatment. Furthermore, Adipsin appears to regulate bone remodeling in non-obese conditions such as CR, fasting, and aging. Thus, Adipsin may contribute to bone loss observed with aging and diet-induced disorders. Clinical observations support this hypothesis given that Adipsin levels are elevated in post-menopausal women with associated low BMD (*Azizieh et al., 2019*) and circulating levels of Adipsin and associated complement proteins fluctuate in response to food intake changes in patients with anorexia nervosa (*Pomeroy et al., 1997*). Since Adipsin is a major regulator of the complement system, current pharmacological advancements include the synthesis and preclinical characterization of Adipsin inhibitors targeting the alternative complement pathway (*Karki et al., 2019*). For example, an anti-Adipsin antibody has been developed for the purpose of treating geographic atrophy (*Kassa et al., 2019*). Thus, based on our work and others there is potential for these existing treatments to be repurposed to treat and prevent age-related skeletal disorders.

## Acknowledgements

We thank Sam Robinson for assistance with techniques in bone analysis and Ana M Flete and Thomas Kolar for technical assistance with animal studies. This work was supported by the National Institutes of Health T32DK007328 (NA), F31DK124926 (NA), R01DK121140 (JCL), R01AR068970 (BZ), R01AR071463 (BZ), R01DK112943 (LQ), R24DK092759 (CJR), and P01HL087123 (LQ).

## Additional information

### Funding

| Funder | Grant reference number | Author |
| --- | --- | --- |
| National Institutes of Health | T32DK007328 | Nicole Aaron |
| National Institutes of Health | F31DK124926 | Nicole Aaron |
| National Institutes of Health | R01DK121140 | James Lo |
| National Institutes of Health | R01AR068970 | Baohong Zhao |
| National Institutes of Health | R01AR071463 | Baohong Zhao |
| National Institutes of Health | R01DK112943 | Li Qiang |
| National Institutes of Health | R24DK092759 | Cliff J Rosen |
| National Institutes of Health | P01HL087123 | Li Qiang |

The funders had no role in study design, data collection and interpretation, or the decision to submit the work for publication.

### Author contributions

Nicole Aaron, Conceptualization, Formal analysis, Funding acquisition, Investigation, Methodology, Writing - original draft, Writing - review and editing; Michael J Kraakman, Conceptualization, Investigation, Writing - review and editing; Qiuzhong Zhou, Formal analysis; Qiongming Liu, Samantha Costa, Jing Yang, Longhua Liu, Lexiang Yu, Liheng Wang, Ying He, Lihong Fan, Investigation; Hiroyuki Hirakawa, Lei Ding, Baohong Zhao, Edward Guo, Methodology; James Lo, Funding acquisition, Methodology, Writing - review and editing; Weidong Wang, Methodology, Writing - review and editing; Lei Sun, Formal analysis, Methodology, Writing - review and editing; Cliff J Rosen, Formal analysis, Writing - review and editing; Li Qiang, Conceptualization, Supervision, Funding acquisition, Project administration, Writing - review and editing

## Author ORCIDs

Lei Ding (iD) http://orcid.org/0000-0003-4869-8877
Lei Sun (iD) https://orcid.org/0000-0003-3937-941X
Li Qiang (iD) https://orcid.org/0000-0001-8322-1797

## Ethics

Animal experimentation: All animal protocols used in this study were reviewed and approved by the Columbia University Animal Care and Utilization Committee (Protocol # AAAX9458).

## Decision letter and Author response

Decision letter https://doi.org/10.7554/eLife.69209.sa1
Author response https://doi.org/10.7554/eLife.69209.sa2

## Additional files

### Supplementary files

• Transparent reporting form

### Data availability

The gene expression dataset of whole tissue bone marrow (BM) in mice fed ad libitum or on calorie restriction for 3 weeks is from GEO (GSE124063).

The following previously published datasets were used:

| Author(s) | Year | Dataset title | Dataset URL | Database and Identifier |
|---|---|---|---|---|
| Collins N, Han S-J, Enamorado M, Link VM, Belkaid Y | 2019 | The Bone Marrow Protects and Optimizes Immunological Memory during Dietary Restriction | https://www.ncbi.nlm.nih.gov/geo/query/acc.cgi?acc=GSE124063 | NCBI Gene Expression Omnibus, GSE124063 |

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
