## [Decision Letter]

**Acceptance summary:**

This study reveals a novel mechanism by which the adipokine adiposin regulates bone marrow adipose tissue expansion and bone mass through paracrine and endocrine actions using a complement-dependent mechanism. Adipocyte secretion of adipsin was found to be physiologically related to calorie restriction, aging and treatment with PPARgamma ligands.

**Decision letter after peer review:**

Thank you for submitting your article "Adipsin promotes bone marrow adiposity by priming mesenchymal stem cells" for consideration by *eLife*. Your article has been reviewed by 3 peer reviewers, and the evaluation has been overseen by a Reviewing Editor and a Senior Editor. The following individuals involved in review of your submission have agreed to reveal their identity: Feng Liu (Reviewer #1); Stephen Farmer (Reviewer #3).

Essential Revisions:

1. Please address the minor concerns of Reviewer 1.

2. Please address the minor concerns of Reviewer 2.

*Reviewer #1 (Recommendations for the authors):*

This manuscript could be further improved if the following questions could be addressed/discussed:

1. What could be the mechanism by which Adipsin deficiency reduced fat mass in BM but increased fat mass in other fat depots?

2. The authors found that C3 knockout had a pronounced inhibitory effect on BMAT expansion but only had a mild effect on bone loss (Figure 3), suggesting an uncoupling of BMAT expansion from bone loss. However, ablation of C3 enhanced osteoblast differentiation capacity. It is unclear whether the effect of C3 deficiency on osteoblast differentiation is associated with BMAT expansion.

3. The authors found that Adipsin inhibits Wnt signaling in BMSCs but had no significant effect on the p-GSK3b and b-Catenin in the eWAT. What could be the mechanism by which Adiposin selectively regulates Wnt signaling in BMSC?

4. Adipsin levels in the eWAT, sWAT, and BAT as well as in circulation were markedly decreased in the PPARr deacetylation-mimetic 2KR mice. Are the levels of Adiposin changed in BMAT of the 2KR mice?

5. The authors presented the data of Figure 1B before 1A, it would be better to switch the order of these two figures.

*Reviewer #2 (Recommendations for the authors):*

1. In order to determine if adipsin's action is dependent on C3, the authors could treat control and C3^-/-^ BMSCs with adipsin, followed by adipogenic and osteogenic differentiation and gene expression analyses.

2. Changes of gene expression were more evident in Adipsin^-/-^ and C3^-/-^

(Figure 5) than Adiposin/SB treatment (Figure 6), suggesting a possible indirect effect of altered BM composition of Adipsin^-/-^ and C3^-/-^. Can the authors perform Adipsin or C3 knockdown in isolated wildtype BMSCs? Would exogenous Adipsin rescue defects of Adipsin^-/-^ BMSCs?

3. The effect of CR (Figure 2A, Figure S2E) and HFD (Figure 3G, Figure 3I) on the expansion of regulated BMA in wildtype mice was not evident. Perilipin IHC is an alternative to quantify BMAT.

4. It is unclear what the three panels of each genotype in Figure 2H refer to. Different mice or locations?

*Reviewer #3 (Recommendations for the authors):*

No recommendations. A well executed set of experiments.

---

## [Author Response]

Reviewer #1 (Recommendations for the authors):This manuscript could be further improved if the following questions could be addressed/discussed:1. What could be the mechanism by which Adipsin deficiency reduced fat mass in BM but increased fat mass in other fat depots?

We suspect that the mechanism by which Adipsin deficiency reduces fat mass in the BM but not in the peripheral fat depots is due to an impairment of lipid mobilization from peripheral fat to bone marrow in the absence of Adipsin, especially during calorie restriction. As we have noted, Adipsin appears to regulate the plasticity of marrow fat in response to various nutrients. Thus, in the state of calorie restriction, the plasticity of marrow fat in Adipsin KO mice is less responsive to store energy relocated from peripheral fat depots. This is further reinforced by our unpublished lipidomic, proteomic, and RNAseq data from our human studies of fasting in which marrow adipocyte expansion is related to enhanced uptake of fatty acids from the circulation. We have added this comment in the Discussion section.

2. The authors found that C3 knockout had a pronounced inhibitory effect on BMAT expansion but only had a mild effect on bone loss (Figure 3), suggesting an uncoupling of BMAT expansion from bone loss. However, ablation of C3 enhanced osteoblast differentiation capacity. It is unclear whether the effect of C3 deficiency on osteoblast differentiation is associated with BMAT expansion.

We agree with the reviewer’s comment. Our data does in fact suggest a potential uncoupling of osteoblasts and adipocytes in the bone marrow, not unlike other mouse models, both in genetic and environmentally induced manners (e.g. short term TZD treatment in Figure 1J-L). We suspect it is more likely that the bone marrow is being primed in development by the presence or absence of Adipsin and C3. Thus, in vitro we see a more striking result. In addition, C3 is involved in many pathways that may be regulated differentially in vivo to overcome the effects on bone loss and minimize the phenotype. We have added a comment to address this concern in the Discussion section. In future work, we plan to use C3a receptor conditional knockout mice to explore this more precise pathway of the complement system. The uncoupling of osteoblasts and BMATs is currently a popular area of interest in the field that we hope to explore further in future studies.

3. The authors found that Adipsin inhibits Wnt signaling in BMSCs but had no significant effect on the p-GSK3b and b-Catenin in the eWAT. What could be the mechanism by which Adiposin selectively regulates Wnt signaling in BMSC?

We speculate that the difference in p-GSK3b and b-Catenin has two possible mechanisms. First, the environments in the BM and the peripheral adipose tissue depots differ in response to various stimuli. It is well understood that Wnt signaling is regulated by multiple factors and that stromal cell differentiation is very dependent on the site of origin. In EWAT, it is conceivable that other pathways affecting Wnt signaling could overcome the effect of Adipsin. Second, it may be related to adipocyte lineage determination. In the bone marrow, the progenitor bone marrow stromal cells (BMSCs) can differentiate into adipocytes or osteoblasts, whereas in EWAT, the progenitor cells are primed toward adipogenesis in this adipocyte-rich environment. In the marrow, Wnt signaling is crucial in determining BMSC fate determination but less important in EWAT where fate is already determined and the differentiation relies more heavily on PPARγ2 expression. For example, we have previously shown that the 2KR mutation did not affect the adipogenic capacity of the PPARγ2 isoform in mouse embryonic fibroblasts (MEFs) but did impair the adipogenic capacity of PPARγ1 (1). Indeed, in our unpublished work, we found impaired adipogenesis in the BMSCs isolated from 2KR mice but not in peripheral adipose tissue BMSCs. We have added a brief discussion on this point.

4. Adipsin levels in the eWAT, sWAT, and BAT as well as in circulation were markedly decreased in the PPARr deacetylation-mimetic 2KR mice. Are the levels of Adiposin changed in BMAT of the 2KR mice?

We have included gene expression analyses from ad libitum fed WT and 2KR mice in which Adipsin levels are decreased specifically in the bone marrow of 2KR mice (Figure 7 —figure supplement 1D).

5. The authors presented the data of Figure 1B before 1A, it would be better to switch the order of these two figures.

We appreciate this point and have switched the order in which the figures appear to make the order flow better.

Reviewer #2 (Recommendations for the authors):1. In order to determine if adipsin's action is dependent on C3, the authors could treat control and C3^-/-^ BMSCs with adipsin, followed by adipogenic and osteogenic differentiation and gene expression analyses.

Due to the better specificity of the C3aR inhibitor, SB290157, we used WT BMSC and treated the cells with SB290157 to mimic inhibition of the alternative complement pathway, specifically. We then treated the cells with or without recombinant mouse Adipsin during osteoblast and adipocyte differentiation. Gene expression revealed no changes in adipogenesis or osteoblastogenesis with exogenous Adipsin when C3aR is inhibited. This further validates our hypothesis that Adipsin affects the bone marrow homeostasis through its role in the alternative complement pathway. The results of this experiment have been added to Figure 6 —figure supplement 1G-H and discussed.

2. Changes of gene expression were more evident in Adipsin^-/-^ and C3^-/-^(Figure 5) than Adiposin/SB treatment (Figure 6), suggesting a possible indirect effect of altered BM composition of Adipsin^-/-^ and C3^-/-^. Can the authors perform Adipsin or C3 knockdown in isolated wildtype BMSCs? Would exogenous Adipsin rescue defects of Adipsin^-/-^ BMSCs?

While we appreciate this suggestion, Adipsin is not highly expressed in WT BMSCs so we predict that the further knockdown of a gene that is expressed at low levels, would not have a strong effect. As previously mentioned, we suspect the in vitro effect is pre-determined by the BM composition priming these cells in vivo, possibly already changing fate determination. This helps to explain why we don’t see as strong of an effect in WT cells. We have treated Adipsin KO BMSCs with exogenous Adipsin (Figure 6B and Figure 6 —figure supplement 1A-C) and observed a reverse phenotype, implying that exogenous Adipsin does rescue the defects of Adipsin KO BMSCs.

3. The effect of CR (Figure 2A, Figure S2E) and HFD (Figure 3G, Figure 3I) on the expansion of regulated BMA in wildtype mice was not evident. Perilipin IHC is an alternative to quantify BMAT.

We thank the reviewer for asking this question because in addressing this question we have reassessed our analyses. The distal femoral region in which we have focused the osmium tetroxide staining is often considered indicative of the regulated marrow adipose tissue (rMAT) and not the constitutive marrow adipose tissue (cMAT), as previously stated. However, the supporting data in the field is inconclusive. Thus, to avoid misinterpretation of the data, we have referred to the adiposity in the bone as marrow adipose tissue (MAT) to encompass the effects of rMAT and cMAT. This change has been made to our figures and manuscript language. Future studies will address the direct effect of Adipsin on each subtype of MAT, as now mentioned in our Discussion section.

We know that regulated MAT is developed later in life and is highly responsive to conditional changes. The mice used for CR and HFD/Rosi in the Adipsin KO models (Figure 2) were younger than the mice used in the C3 KO CR experiment (Figure 3). It is likely that the absence of fully, expected adipose expansion in the BM is due to the younger age of these mice. In the C3 KO CR model we used older mice and saw a greater expansion of adiposity during CR.

We appreciate the suggestion to analyze perilipin, as it is a robust gene in response to changes in BM adiposity. We did observe significant upregulation of Perilipin in the BM of CR mice as compared to ad. libitum chow fed mice (Figure 1C) and in the BM of HFD and Rosi treated mice as compared to HFD only controls (Figure 1G). In the Adipsin KO and C3 KO mouse models that displayed a decrease in MAT by osmium tetroxide staining, we found a concurrent decrease in expression of Perilipin in the BM as compared to WT mice (shown in Author response image 1), further supporting our hypothesis.

**Author response image 1. sa2fig1:** 

4. It is unclear what the three panels of each genotype in Figure 2H refer to. Different mice or locations?

These panels are from different mice taken in approximately the same location on the bone. This has been clarified in the figure description.

References

1. Kraakman MJ et al. PPAR? deacetylation dissociates thiazolidinedione’s metabolic benefits from its adverse effects. *J. Clin. Invest.* [published online ahead of print: 2018]; doi:10.1172/JCI98709

2. Ding L, Saunders TL, Enikolopov G, Morrison SJ. Endothelial and perivascular cells maintain haematopoietic stem cells [Internet]. *Nature* 2012;481(7382):457–462.

3. Zhong L et al. Single cell transcriptomics identifies a unique adipose lineage cell population that regulates bone marrow environment. e*Life* 2020;9. doi:10.7554/*eLife*.54695